psychology/cognition

facial expression, avoidance learning, electromyography, decision-making, response time

**Author for correspondence:**
Jonathan Yi
e-mail: jonathan.yi@ki.se

# The face value of feedback: facial behaviour is shaped by goals and punishments during interaction with dynamic faces

## Jonathan Yi[1], Philip Pärnamets[1,2] and Andreas Olsson[1]

[1]Department of Clinical Neuroscience, Division of Psychology, Karolinska Institutet, Solna, Sweden
[2]Department of Psychology, New York University, New York, NY, USA

JY, 0000-0001-6624-695X; PP, 0000-0001-8360-9097

Responding appropriately to others' facial expressions is key to successful social functioning. Despite the large body of work on face perception and spontaneous responses to static faces, little is known about responses to faces in dynamic, naturalistic situations, and no study has investigated how goal directed responses to faces are influenced by learning during dyadic interactions. To experimentally model such situations, we developed a novel method based on online integration of electromyography signals from the participants' face (corrugator supercilii and zygomaticus major) during facial expression exchange with dynamic faces displaying happy and angry facial expressions. Fifty-eight participants learned by trial-and-error to avoid receiving aversive stimulation by either reciprocate (congruently) or respond opposite (incongruently) to the expression of the target face. Our results validated our method, showing that participants learned to optimize their facial behaviour, and replicated earlier findings of faster and more accurate responses in congruent versus incongruent conditions. Moreover, participants performed better on trials when confronted with smiling, when compared with frowning, faces, suggesting it might be easier to adapt facial responses to positively associated expressions. Finally, we applied drift diffusion and reinforcement learning models to provide a mechanistic explanation for our findings which helped clarifying the underlying decision-making processes of our experimental manipulation. Our results introduce a new method to study learning and decision-making in facial expression exchange, in which there is a need to gradually adapt facial expression selection to both social and non-social reinforcements.

# 1. Introduction

Although the perception and responses to static faces have been studied in considerable detail (e.g. [1–4]), researchers have only recently begun to use dynamic face stimuli to examine the temporal dimension of the perception and responses to facial stimuli [5–9]. Surprisingly, no study has attempted to experimentally model the learning process of optimizing individuals' goal-directed facial responses during facial expression exchange with a target interactant. In such situations, observers' own facial responses are expected to be reinforced by both the intrinsic value of the interaction partner's facial expression (e.g. smiling and frowning), as well as the outcome of the interaction, for example, if it results in an aversive consequence or not. Our novel paradigm enabled us to study how participants learned to optimize their facial behaviours as a function of exposure to specific facial expressions (smiles and frowns) and their consequences (the absence and presence of aversive stimulation).

Everyday social life presents a host of challenges that can be addressed by adaptively changing behaviour based on past experience. For example, if another individual suddenly approaches you with an angry facial expression, how you respond is likely to be critical for the outcome of the encounter. You may take a confrontational approach and reciprocate the angry expression, thus risking conflict, or take a more submissive approach by smiling, and thereby increase the chances of avoiding harm. If the latter approach is chosen and the other person walks by, you learn that the best course of action to avoid danger from that individual may be to smile whenever he or she exhibits hostility toward you. Learning to actively update and flexibly adjust one's social, communicative behaviour is key to remaining alive and safe [5,10,11]. Therefore, an important question is to determine what underlying mechanisms influence the learning process of our facial expression selection during interactions in a threatening environment. To address this question, we developed a novel method based on online integration of electromyography (EMG) signals from the participants' face during the confrontation with other target interactants (facial stimuli) displaying dynamic facial expressions. The possible aversive consequence of choosing the 'wrong' expression was modelled by the looming threat of receiving mild electrical stimulation as a function of smiling and frowning towards the other target interactants.

Facial expressions sometimes involuntarily communicate emotional states. For example, past research has shown that spontaneous facial mimicry facilitates congruent exchanges of facial expressions [12], and that top-down cognitive control is important to suppress facial mimicry [13–15]. Facial expressions can also be used deliberately in a goal-oriented communication [5,16–18], and successful communication of emotional states can prevent potentially harmful encounters [19]. In naturalistic interactions, automatic and goal-directed responses will at times be aligned and at other times be in conflict, such as when an individual learns to frown rather than to smile towards a potential bully on the school yard or at the work-place to stand up for themselves.

## 1.1. Smiles and frowns function as reinforcing stimuli

Facial expressions are controlled by several independent striated muscles that are under voluntary control. These muscles can be activated independently or in synchrony with each other to form distinct facial expressions and transmit information about, for example, emotional states and intentions [20]. In social interactions, a smiling face can reinforce specific behaviours in the perceiver through its rewarding qualities [17,21,22]. Additionally, a previous study showed that in a decision-making task, participants displayed increased propensity to repeat actions reinforced with genuine smile feedback compared with a non-social feedback [23]. At the same time, an angry face can serve as a punisher, thus attenuating behaviours, facilitate avoidance [24] and increase autonomic arousal [25,26]. Furthermore, angry expressions have been demonstrated to curtail the behaviour of conspecifics in cases where social norms were violated [27]. In sum, there is considerable evidence that the human face functions both communicatively and as a primary reinforcer.

## 1.2. Instrumental avoidance learning

Avoidance learning is a form of instrumental learning [28] describing how the likelihood of performing certain behaviours is diminished through experiencing stressful or unpleasant events [28,29] and is naturally present in social interaction [30]. Facial expressions play a central role in social interaction [5,16,17,31,32], and avoidance learning is likely to be involved in the optimization of facial expression

selection and adaptation over time during interactive dyads. Given the ubiquity of avoidance learning to calibrate real-life social interactions, we designed a task where participants smiled and frowned towards two smiling or frowning target interactants (one for each expression, respectively). For each unique individual target interactant, participants learned through trial-and-error to reciprocate or respond with the opposing facial expression. Behaving suboptimally or 'wrongly' entailed the possibility to receive an aversive shock.

Our social environment is in constant flux, and failing to adjust to changing reinforcement contingencies can have detrimental effects on our ability to effectively navigate our social world [33]. For example, not realizing that an interlocutor's smile signalling friendliness suddenly changed to an angry frown signalling hostility when engaging in casual banter can lead to social gaffes or damaged relationships. Moreover, norms that govern behaviour dynamically change across social contexts, and hence, it is important to adjust one's behaviour, for example, facial expressions, to avoid aversive consequences. To model this variable aspect of our social life: how the facial expressions and their motivational meaning can change during the course of interaction, we introduced a change of contingencies (reversal learning) halfway through the experimental procedure. A second reason for introducing reversal learning was to provide improved opportunities to measure how participants learned and relearned the contingencies. For example, the pattern of choices immediately following the applied analytic models to the learning and decision-making processes. In summary, we reversed the contingencies between target expression (expression of the target interactant) and optimal facial action after the first half of the trials. This meant that if the correct response (CR) to a frown by a particular target interactant was to smile during the first half of the experiment, the optimal response was to frown back during the latter half.

Furthermore, instrumental and avoidance learning can be formalized through computational models. Here we used reinforcement learning (RL) to model how participants adapted to the demands of our task. The goal of a learner under RL is to maximize the net rewards they receive while interacting with a complex, uncertain environment [34,35]. This is modelled by assuming that learners represent the expected value of responses available to them and probabilistically select between responses on each trial. These internally represented expected values are updated when the outcomes of actions are observed. The form updating takes is determined by the learning rule. In this work, we assumed the Rescorla–Wagner learning rule [34], which assumes that learners update values by a fraction of the difference between their expectation and the outcome. RL models are increasingly applied to understand adaptive behaviour in both social and non-social tasks, as well as to link adaptive behaviour with its neural basis [10,36]. In the present study, participants learned through trial and error which facial responses were correct as a function of the individual target interactant that they interacted. We evaluated several competing RL models of participants' learning process, allowing us to characterize how participants adapted to the task including the reversal of reinforcement contingencies halfway through the experiment. Specifically, we compared a baseline model where participants' learned only through trial-and-error with several alternative specifications where participants priors, decision-making, or learning were biased by tendencies to copy interactants or by what expressions (smiles or frowns) interactants made. Using RL models allowed us to begin to build towards the aim of understanding the algorithms participants use when learning about and adapting to facial expression contingencies.

## 1.3. Drift diffusion model

In the present study, participants learned to minimize punishments by reciprocating and responding incongruently towards the target interactant. Since humans often mimic the facial expressions of others, participants might be predisposed, or biased, when deciding what facial expression to form, towards reciprocating the target expression of their target interactants [2,12]. In the present study, participants also learned through trial and error which facial responses were correct to what individual target interactant that they interacted with. In order to disentangle the mechanisms underlying participants' decision-making processed during this learning process, we used a drift diffusion model (DDM) [37] to model the underlying evidence accumulation leading up to selecting one facial expression over another. The DDM is part of a class of sequential sampling models developed to account for the importance of response times (RT) during binary decision-making in perceptual discrimination tasks [37]. These models assume that evidence for a specific response is accumulated stochastically and that the efficiency of this process is reflected in the amount of evidence available for that response. The drift diffusion model describes this through a drift rate parameter which gives the slope of the average rate of accumulation (V) (figure 1). Higher drift rate

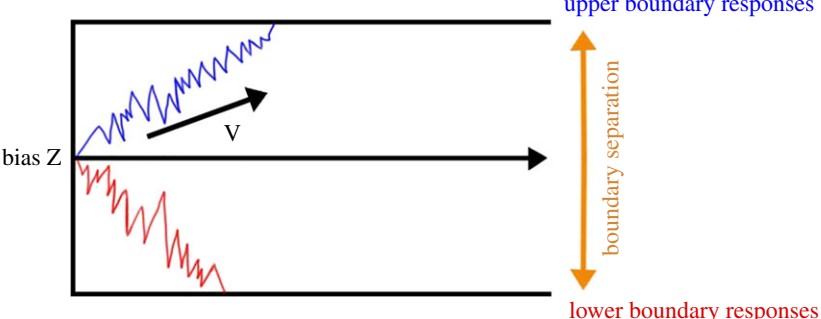

**Figure 1.** Evidence for a specific response is accumulated stochastically and the velocity of this process is reflected in the drift rate (V). Boundary separation represents the threshold for evidentiary weight that is required to arrive at a decision and hence quantifies the speed–accuracy trade-off. Furthermore, bias (Z) refers to the bias of the starting point where evidence is being accumulated towards a specific decision.

means greater evidence accumulation per time unit towards a specific decision: if the drift is positive, reaching the upper boundary response is more likely, and if the drift is negative, reaching the lower boundary is more likely. In our framework, we modelled the upper response as participants' forming a 'smile' and the lower as participants' forming a 'frown'. In the drift diffusion model, boundary separation represents the threshold for evidentiary weight that is required to arrive at a decision and hence quantifies the speed–accuracy trade-off. High values of boundary separation increase accuracy and lead to slower RTs, whereas low values result in quick decisions, which increases error rate. The drift diffusion model also contains a bias parameter (Z, figure 1), allowing the evidence accumulation to begin at an offset from the equidistant middle point. To the best of our knowledge, no study has previously applied the drift diffusion model to study decisions controlling facial muscular activity. By applying the drift diffusion model to our study, we expected to gain a better understanding of the influences of various sources of information (facial expression and experimental condition) on participants' behaviour and how these influence evidence accumulation and speed–accuracy trade-off in facial expression selection.

## 1.4. The current study

To address the question of how we gradually learn to optimize our facial expressions in interactive dyads, we introduced a novel experimental model which simulated facial expression exchange between participants and target interactants (dynamic facial stimuli). This novel experimental model and method integrated participants' own facial EMG signals, and based on these signals, provides reinforcing biofeedback in the form of mild electrical stimulation. The participants interacted with dynamic faces that either smiled or frowned at them, and learned to avoid shocks through trial-and-error by expressing the same (congruent) or different (incongruent) expression to the presented face. We measured the participants' CR and RT for smiling and frowning.

An important objective was to establish and test a new experimental model to study learning and decision-making in interactive dyads and to investigate how these processes were dependent on facial expression (smiling or frowning) of the interaction partner, as well as the goal to reciprocate or oppose the target interactant's expression. We postulated four specific hypotheses: (i) Improved adaptive learning: We expected CR to increase. (ii) Congruency facilitates learning: We hypothesized that participants would be (iia) more accurate and (iib) respond faster during learning the contingencies when the CR was to reciprocate (congruent trials) as compared with when it was not (incongruent trials), because spontaneous facial mimicry facilitates congruent exchanges of facial expressions [2,12], and because congruent responding to visually displayed motor actions is thought to be facilitated by cortical mirror neuron activation along the dorsal visual stream [38,39]. (iii) Smiling target interactants facilitate learning: We anticipated to observe more CR for trials where the target interactant that the participant was interacting with was smiling compared with when the target interactant was frowning. This hypothesis was prompted by simulations based on an earlier pilot study (see Methods, Participants). (iv) Reversal learning: We explored if there would be a difference in learning new contingencies as a function of the order of expressions (frowning versus smiling). In other words, we investigated whether it would be easier to optimize facial expression by smiling after having frowned or the other way around. Specifically, we investigated if participants would perform better or worse

|  | congruent | incongruent |
|---|---|---|
| frown | A. Frown to frowning face to avoid shock. | C. Smile to frowning face to avoid shock. |
| smile | A. Smile to smiling face to avoid shock. | D. Frown to smiling face to avoid shock. |

**Figure 2.** A design matrix showing the four different conditions (all within-subjects) of the experimental manipulation.

before versus after reversal as a function of congruent responding to facial expressions as well as the target expression of the target interactant. We expected that participants would have fewer CR following the reversal in early trials compared with late trials. Finally, we added DDM and RL models to shed light on the computational mechanisms underlying the decision-making and learning processes contributing to how participants adapt to the experimental contingencies they faced. Nevertheless, the modelling was primarily exploratory and secondary to the main goal of the present work, namely to establish our method for studying adaptive facial responses.

## 2. Methods

### 2.1. Participants

All participants were recruited at the Karolinska Institutet (Stockholm, Sweden) campus and provided written consent. All participants provided a written and informed consent prior to participation. The present study was approved by the Regional Ethic Committee of Stockholm (Dnr: 2017/37–31/4). Participants received two cinema vouchers as compensation. We recruited 71 participants (30 female) who were compensated for their participation. Participants were volunteers and they were not selected based on any gender- or ethnicity-based criteria. For reasons related to our ethical review, participant ethnicity was not asked or otherwise noted. Thirteen participants were excluded due to low EMG signals, which were not detected by the software, thus interrupting the biofeedback procedure. The final sample consisted of 58 participants (29 female, mean age = 26.4, s.d. = 4.3). The sample size was targeted at 60 and it was determined using a power simulation method described in [40] and we found a 90% chance to find significant main effects of Congruency and Expression when using CR as the dependent variable. A design matrix showing all the four different conditions of our experimental manipulation is illustrated in figure 2.

### 2.2. Materials

#### 2.2.1. Stimuli

Twelve video clips were retrieved from the Amsterdam Dynamic Facial Expression Set (ADFES) [9]. Six of the video clips consisted of video presentations of the faces of three different male individuals. Each male individual appeared in two video clips, expressing a smile and a frown, respectively. The other six video clips consisted of video clips of the faces of three different female individuals. Each female appeared in two video clips, expressing a smile and a frown, respectively. The sex of the target faces was matched to the participants' genders in order to minimize any potential confounding effects due to inter-gender interaction. All of the individuals presented in the video clips were of northern European descent. Eight of the 12 video clips were used in the experimental manipulation (two male and two female faces) while the remaining four video clips (one male and one female face), were used in the shock calibration and practice phase. All the smiling facial stimuli demonstrated major activation of Action unit (AU) 12CDE (zygomaticus major) while all frowning facial stimuli demonstrated major activation of AU 4CDE (corrugator supercilii) according to the Facial Action Coding System [41]. The unconditioned stimulus consisted of a 200 ms DC-pulse electric stimulation (administered using the STM200; BIOPAC Systems) that was applied to the participant's right volar forearm.

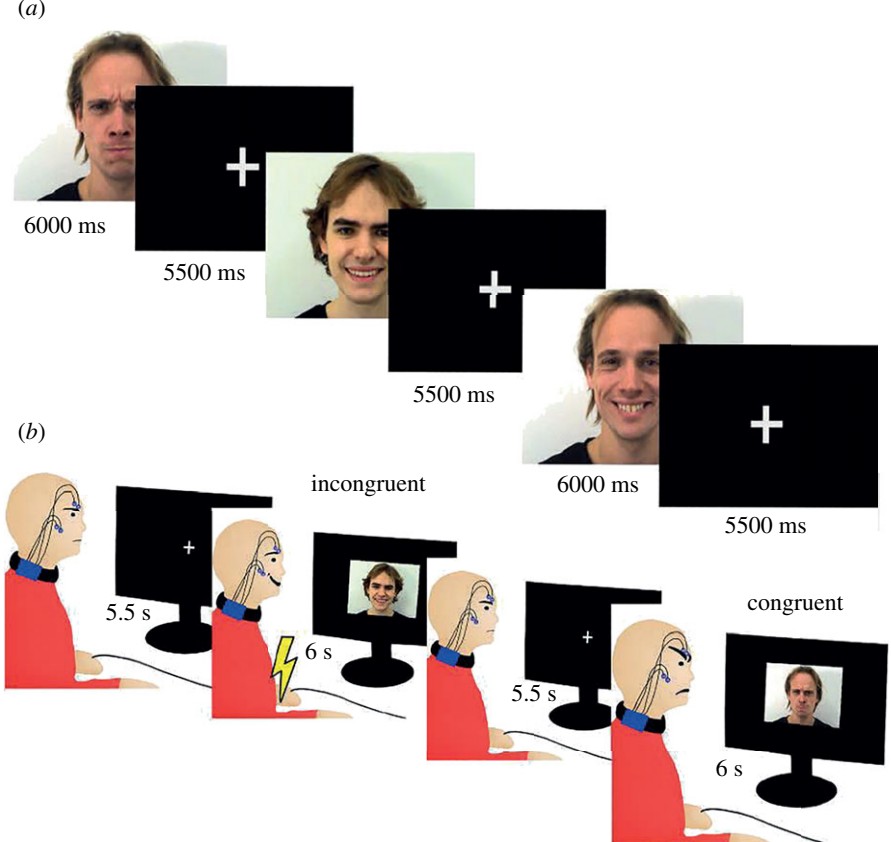

**Figure 3.** (*a*) Presentation structure of stimuli material in the experimental manipulation. 6000 ms video clips of faces (faces of target interactants) forming either happy or angry expressions were presented and superseded by a 5500 ms inter-stimulus interval (ISI). (*b*) One of the target interactants was assigned the congruent condition while the other target interactant was assigned the incongruent condition. In this avoidance learning task, the participants learned by trial-and-error to avoid mild electric shocks by expressing the same (congruent) or different (incongruent) expression in response to the expression of the target interactant. Each target interactant was assigned to the congruent or incongruent condition. Each video clip of a target interactant was presented for a total of 6000 ms where in the initial 500 ms, the target face displayed a neutral expression and subsequently formed either a happy smile or an angry frown for 5.5 s. Each video presentation trial was superseded by an ISI.

## 2.2.2. Data acquisition

We applied 4 mm AgCl surface electrodes on the corrugator supercilii and zygomaticus major muscle groups respectively on the left side of the participants' faces using a bipolar configuration [42]. One electrode was then attached to top of the midsection of the forehead beneath the hairline for grounding the recorded signal. An EMG100c (BIOPAC SYSTEMS) module was used to amplify the signal and forward it to the Acqknowledge 4.3 software system (BIOPAC SYSTEMS). Raw EMG signals from corrugator supercilii and zygomaticus major were averaged over 30 samples at 1.0 kHz.

## 2.3. Procedure

### 2.3.1. Shock calibration and practice phase

A work-up procedure was used to calibrate the shock-level for each participant to find a level that was subjectively experienced as uncomfortable, but not painful [43–45].

The participants first engaged in a practice session in order to prepare for the main experiment where they were presented with two 6000 ms video clips of a face that either smiled or frowned upon presentation (figure 3). Every video presentation trial was followed by a 5500 ms inter-stimulus interval (ISI). In conjunction with video clip presentation, written instructions were displayed on the same monitor that instructed the participants to either smile or frown.

## 2.3.2. Experimental manipulation

The experiment entailed repeated presentation of video clips of two different target interactants that either smiled or frowned angrily upon presentation (see figure 2 for an overview of the experimental design). Only one video clip was shown at a given trial. Before the start of the experiment, the participants were instructed to either smile or frown when the video clips were presented. Mild aversive stimulation was administered if the participant formed the 'incorrect' facial expression. Figure 3 illustrates presentation structure of the stimuli material in the experimental manipulation.

We used four different video clips consisting of two different faces (faces of the target interactants) forming two different facial expressions (either a smile or a frown). The participant's task was to learn through trial and error to form the 'correct' facial expression for each video clip and if the 'incorrect' facial expression was formed by the participants, he or she received a mild electrical shock to his or her wrist. Each video clip had a correct facial response assigned to it in terms of the participant either smiling or frowning toward it during presentation of the video clips. The participants interacted with two different target interactants and they had to consistently match and reciprocate the facial expression of one of the target interactants (target interactant 1) and conversely, respond with the opposite expression to the other target interactant (target interactant 2). For example, if target interactant 1 smiled, the participant's task was to reciprocate the smile and vice versa when target interactant 1 frowned. For the other target interactant (target interactant 2), the participant's task was to always form the opposite expression of the one being shown by the target interactant. For example, if a smile was being formed by target interactant 2, the participant's task was to respond with a frown and vice versa for when target interactant 2 smiled (see figure 3 for some examples). If the participant did not form either facial expression, aversive stimulation was delivered at the end of the video clip. If the correct facial expression was formed according to these contingencies, the participant avoided aversive stimulation during the first half of the experiment. To clarify, participants were merely instructed to learn through trial and error to form an expression during each trial to avoid electrical shocks as function of which target interactant that they were interacting with as well as the expression that the target interactant formed. In other words, the participants were never informed of what constituted a correct or incorrect facial response. The full instructions given to the participants can be found in the electronic supplementary material (p. 6).

The presentation order of the video clips in the experiment was randomized with the restriction that the same video clip could not be shown more than two times consecutively (for example a video clip of target interactant 1 smiling could not be displayed three or more trials in a row etc.). In the first half of the experiment, each of the four video clips (video clips of target interactant 1 and target interactant 2 smiling) were shown 12 times, resulting in a total of 48 trials. Following this first phase, a second phase immediately ensued without notifying the participants. In this second phase, the aversive stimulation conditions were reversed, meaning that the CR to target interactant 1 and target interactant 2 were the reverse of the first half of the experiment. This second phase was otherwise identical to the first phase in terms of stimulus material, ISI and number of trials. In total, the experiment consisted of 96 trials. All conditions were fully within-subjects.

## 2.4. Individual difference measures

For exploratory and control purposes, we included a short battery of tests (after the completion of the experimental manipulation), measuring individual variation in working memory, facial perception, state and trait anxiety as well as emotional regulation. Accordingly, we used an N-back task, an online test of identification of emotional expression, and an assessment of anxiety and self-rated emotional regulation. These individual measures were used for control purposes. However, the limited sample size of our study precluded us from drawing any general conclusions about these traits.

### 2.4.1. N-back task

The participants underwent a 2-back version of the N-back task. The N-back task is widely used to measure working memory capacity [46]. The participants were informed regarding the nature of this task through verbal explanations as well as examples. In total, three blocks of 39 trials were presented. Each trial consisted of a number being presented for 500 ms followed by a 1000 ms ISI. Of the numbers shown, 33% were targets; in other words, they were numbers that were presented two steps back into the sequence. In total, there were nine different numbers that were randomly

presented throughout the task. If the number shown was a target, the participants were supposed to press the left arrow key. Otherwise, if it was not a target, then their task was to press the right arrow key.

### 2.4.2. Facial emotion identification task

The participants underwent a RT test where they were presented with pictures of faces either smiling or frowning upon presentation. The stimuli material consisted of 60 pictures of 30 different individuals, each person forming either happy and angry expressions. The facial stimuli were retrieved from NimStim and KDEF [47,48]. The order of presentation of trials was randomized. Furthermore, the stimuli were gender matched to the participant. The participant's task was to assess whether the faces were smiling or frowning by pressing either left or right arrow key as fast as possible. To clarify, we did not use the same stimuli during this task as in the experimental manipulation

### 2.4.3. Questionnaires

The participants completed the emotion regulation quotient (ERQ) [49] and State-trait anxiety inventory (STAI) [50] questionnaires at the end of the experiment. STAI was used to control for potential differences in state and trait anxiety, and ERQ was used to test whether emotional regulation was a relevant predictor for performance in this task on an exploratory basis. However, as mentioned above, the limited sample size of our study precluded us to draw any general conclusions about these traits.

## 2.5. Data analysis

### 2.5.1. Dependent measures

A facial response was defined as when either the activity in the corrugator supercilii or zygomaticus major reached $0.001 \, \text{mV s}^{-1}$ (averaged over 30 samples at 1 kHz). If this threshold was reached, the participant either avoided or received a shock. The response variable was scored as either 1 (for CR) or 0 (for incorrect responses) for each trial. The RT was defined as the difference in time in seconds from stimulus onset until the participants reached the $0.001 \, \text{mV s}^{-1}$ threshold for either the zygomaticus major or corrugator supercilii muscular activity.

In all analyses, trials with an RT below 626 ms were excluded (618 out of 5568 trials) because these were judged to indicate non-compliance with the task. The choice of cut-off time was based on the timing parameters of our dynamic faces, which held a neutral expression for 500 ms before forming either a frown or a smile. Mimicry responses can be observed already at 126 ms in response to facial stimuli forming facial expressions [51]; hence, we concluded that the fastest possible response to the expression of the target interactant could manifest at 626 ms. Out of these 618 non-compliance trials, 151–159 trials were excluded trials for each combination of trial conditions (incongruent angry, congruent angry and so on), meaning that approximately equally number of trials were excluded for the different condition (incongruent angry, congruent angry etc.). It is likely that most trial exclusions were caused by participants accidently triggering the EMG apparatus, for example, through involuntary twitches of their facial muscles.

Additionally, trials with an RT above 5 s were considered outliers and removed (only one trial). Finally, we removed trials where participants did not form a detectable facial expression in accord with our EMG biofeedback system (45 trials). In sum, we excluded a total 665 trials from a total of 5568 trials resulting in the final sample of 4903 trials.

### 2.5.2. Mixed model regression

To account for repeated measures, missing data points and the structure of our random effects, we used R [52] and lme4 [53] to perform generalized linear mixed models to analyse the participants' CR in the main task (i.e. where each trial was scored as either 0 or 1). We adopted a significance level of $p < 0.05$. Linear mixed models were used to analyse the participants' log-transformed RT. We adopted a stepwise model selection by initially including the total effect of our fixed effects and all their possible interactions and we then removed fixed effects by comparing the $p$-values of the fixed effects and Bayesian information criterion (BIC) values. These values were obtained using the packages *lme4* [53] and *lmertest* [54] in R. Furthermore, if models did not converge, the selection of those models was aborted. Specific model selection can be found in the electronic supplementary material, including

results of using individual difference measures, N-back task and facial identification task as fixed effects (electronic supplementary material, tables S3–S15).

### 2.5.3. Reversal and descriptive statistics

To better understand the interaction between reversal and congruency as observed in our mixed effects model using CR as a dependent variable, we compared participants' CR on early trials compared with late trials in each block (i.e. we compared with the first three trials in block 1 to the first three trials in block 2, followed by an analysis where we compared the last three trials in block one to the last three in block 2).

### 2.5.4. Reinforcement learning

We estimated participants trial-by-trial learning using RL models. Below we outline the winning model, presented in the main text, while alternative models are presented in the electronic supplementary material (pp. 9–10).

The winning model was a simple Q-learning model using a Rescorla–Wagner learning rule. The model assumed that participants, on each trial, learned the value of two actions—copy or not copy (respond congruently or incongruently) separately for each of the two target interactants they faced (since the participants could make two expressions in our task and the target interactants always made one of the same two expressions)

$$Q = \{\text{copy}, \neg\text{copy}\}$$

Participants chose between each action using a standard SoftMax decision rule. In addition, a free parameter $\zeta$ was added to the value of copying, modelling a potential bias towards (or against) copying the facial expression of the target interactant. If $\zeta > 0$ there is a bias towards copying and if $\zeta < 0$ there is a bias towards not copying

$$P(Q_{\text{copy}} \,|\text{Interactant}) = \frac{e^{(Q_{\text{copy}}+\zeta)/\beta}}{e^{(Q_{\text{copy}}+\zeta)/\beta} + e^{Q_{\neg\text{copy}}/\beta}}.$$

Where $\beta$ is a temperature parameter governing the stochasticity of choice. If $\beta < 1$ the value difference between the two actions is amplified while if $\beta > 1$ the value difference is attuned, effectively making choices more random with respect to prior learning. Following a choice, participants receive reinforcement ($r$), −1 for shock and 1 for no shock and calculate a prediction error ($\delta$)

$$\partial_t = r_t - Q_t.$$

The value of the chosen action is then updated, where $\alpha$ is the learning rate,

$$Q_{t+1} = Q_t + \alpha\delta_t.$$

This and competing models were implemented in the Stan [55] probabilistic programming language and fit using MCMC sampling. Models were fit hierarchically to each participant as deviations from an estimated population average. Standard normal priors were applied to all parameters. Learning rate parameters ($\alpha$) were constrained in the interval [0,1] and temperature parameters ($\beta$) were constrained in (0,2), while $\zeta$ was unconstrained. All $Q$-values were initialized to 0.

### 2.5.5. Drift diffusion model

We estimated a hierarchical Wiener diffusion model [56,57] to estimate the joint effects of the experimental manipulation on responses and RT. We used a response-coded model, where the upper boundary response was defined as 'smile' and the lower boundary response was defined as 'frown'. This entails that a positive drift rate in one condition means evidence accumulating for making a smile and conversely for a negative drift rate. We allowed the drift rate and boundary separation to vary as a function of both the target expression and the required expression of the participant on that trial as well as their interaction. The starting point (bias) was allowed to vary depending on the target expression. We allowed all parameters, both intercepts and slopes to vary by participant.

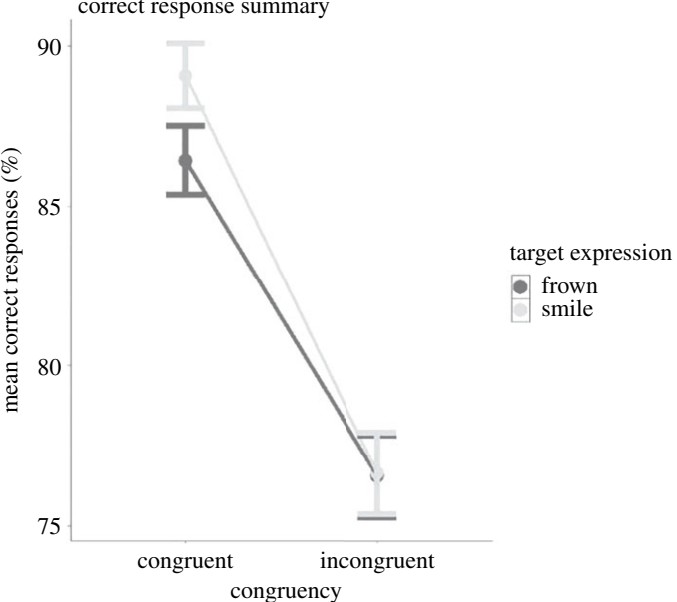

**Figure 4.** Average correct response as a function of Congruency and Expression. Participants performed on average better in the congruent condition. Note: target expression refers to the facial expression formed by the target interactant. Error bars represent standard error (s.e.).

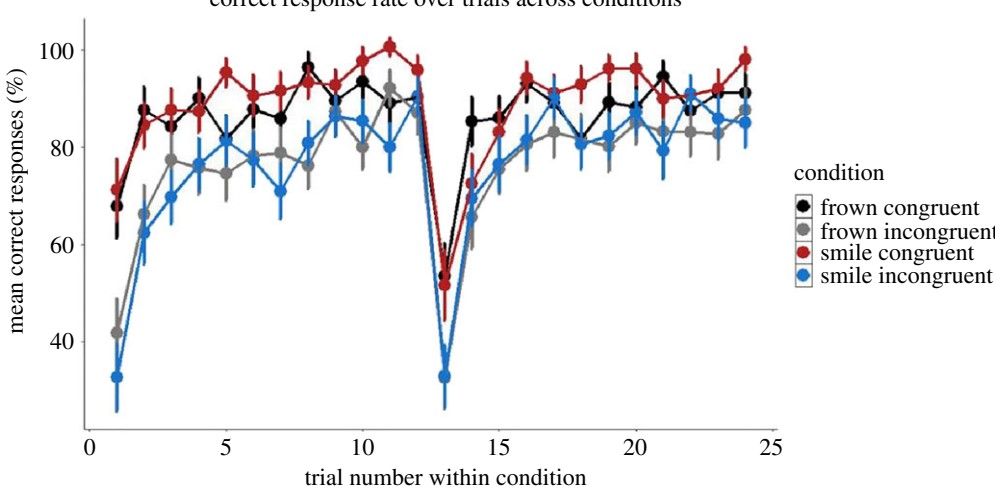

**Figure 5.** Average response times (seconds) as a function of Congruency and Expression. Participants performed overall faster on congruent trials. Note: target expression refers to the facial expression formed by the target interactant. Error bars represent standard error (s.e.). *Y*-axis is truncated.

The model was fitted using *brms* package [55] in R, allowing full Bayesian estimation of the parameters. We ran six chains, each one of them consisting of 10 000 samples with a warm-up of 1000 samples. The priors are shown in the electronic supplementary material (p. 8) and the convergence parameters.

# 3. Results

## 3.1. Correct response

CR was used as a dependent variable including the fixed effects congruency, expression (referring to the expression of the target interactant), trial and reversal and all the fixed effects were standardized around 0. The results of this analysis are summarized in electronic supplementary material, table S1. Graphical summaries of mean CR as a function of Congruency × Expression are illustrated in figures 4 and 5.

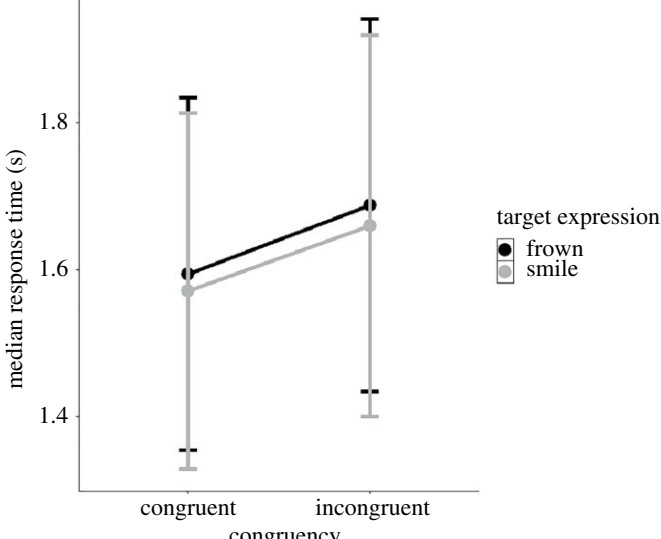

**Figure 6.** Average response times (seconds) as a function of Congruency and Expression. Participants performed overall faster on congruent trials. Note: target expression refers to the facial expression formed by the target interactant. Error bars represent standard error (s.e.). *Y*-axis is truncated.

As predicted, we observed significant main effects of Expression ($B = 1.37$, s.e. $= 0.12$, $p < 0.01$) and Congruency ($B = 2.60$, s.e. $= 0.30$, $p < 0.001$) showing that participants performed better on trials where the target interactant smiled versus frowned as well as on congruent versus incongruent trials.

We also observed a main effect of Trial (relative trial number for each condition) ($B = 31.51$, s.e. $= 0.37$, $p < 0.001$). Additionally, we observed a Congruency × Reversal interaction ($B = 0.57$, s.e. $= 0.21$, $p < 0.01$) showing that participants performed better on average on Congruent trials versus Incongruent trials after the contingency reversal (that is they performed better on trial 13–24 compared with trial 1–12 on congruent versus incongruent trials). We further explore these reversal results below.

## 3.2. Reversal

To better understand if the effects of reversal we observed lasted throughout the second block or were driven by initial difference adjusting to the change in experimental contingencies, we compared performance on the first three trials of each block and on the final three trials on each block. As would be expected given our interaction effect, participants, CR in the first three trials on the first block before reversal were higher compared with trials following the reversal in the congruent condition (80 versus 71%, respectively), with the corresponding numbers in the incongruent condition being 59 and 58%. These differences between blocks diminished in the final three trials with 94% correct prior to reversal and 91% CR after the reversal in the congruent condition and 86% correct both prior to and post reversal in the incongruent condition. However, no differences reached statistical significance (paired *t*-tests, all *p*s greater than 0.17, see electronic supplementary material, tables S16–S19 for details). While inconclusive, we summarize that reversal probably affects participants early in a block, but participants quickly adjust to the new experimental contingencies.

## 3.3. Response time

We conducted a linear mixed model regression testing for the same fixed effects as the CR analysis with the exceptions of replacing CR with RT as the dependent variable. A graphical summary of mean RT as a function of congruency and expression is shown in figures 6 and 7 (raw data and not model fit). We observed significant main effects of Expression ($B = -8 \times 10^{-3}$, s.e. $= 3.8 \times 10^{-3}$, $p < 0.05$), Congruency ($B = -0.02$, s.e. $= 2.8 \times 10^{-3}$, $p < 0.001$) and Reversal ($B = -0.02$, s.e. $= 7.3 \times 10^{-3}$, $p < 0.05$). In other words, participants responded faster on trials where the target interactant frowned and on congruent trials. Furthermore, we observed significant interactions between Reversal × Trial ($B = 0.02$, s.e. $= 6.7 \times 10^{-3}$, $p < 0.01$) where participants displayed slower responses after reversal as trial number increases.

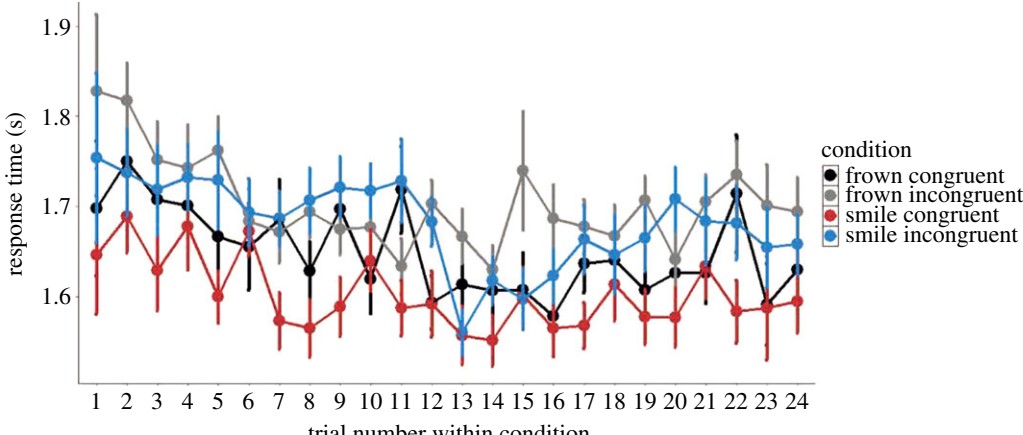

**Figure 7.** Average response times (seconds) showing all four conditions as a function of trial. Trial 13 is the first of each condition after reversal. No effect of reversal was observed. Note: expressions frown and smile in the legend refer to the target expressions formed by the target interactant. Error bars represent standard error (s.e.). Y-axis is truncated.

Finally, we observed an Expression × Congruency × Trial interaction ($B = -0.03$, s.e. = 0.01, $p < 0.05$). The full results of this analysis can be found in electronic supplementary material, table S2.

## 3.4. Correct response and individual differences measures

We conducted six separate logistic mixed models using CR (one model for each individual difference measure) as the dependent variable. In all models, we included interaction effects of Congruency and Expression for fixed effects. Then for each of the models we added an individual difference measure consisting of either STAI-T, STAI-S or ERQ (expression suppression facet and cognitive reappraisal facet), N-back task, facial emotion identification task. (For example, in one model we used Congruency, Expression, STAI-T as fixed effects including the main effects and all possible interaction effects. The next model included Congruency × Expression × STAI-S and so forth.) We observed a main effect of STAI-T ($B = 0.75$, s.e. = 0.14, $p = 0.034$) where increased STAI-T predicted higher CR. The full table of the mixed models can be found in the electronic supplementary material, tables S3–S8.

## 3.5. Response time and individual differences measures

Here conducted six separate linear mixed models using RT as the dependent variable (one model for each individual difference measure). In all models, we included interaction effects of Congruency and Expression for fixed effects. In each model, we added an individual difference measure consisting of either STAI-T, STAI-S or ERQ (expression suppression facet and cognitive reappraisal facet), N-back task, facial emotion identification task as fixed effects. (For example, in one model, we used Congruency, Expression, STAI-T as fixed effects including the main effects and all possible interaction effects. The next model included Congruency × Expression × STAI-S and so forth.) In the facial emotion identification task, we observed a main effect of the RT to detect smiling faces ($B = 0.39$, s.e. = 0.16, $p = 0.02$) where slower detection of smiling faces in the facial emotion identification task predicted slower RT in the experimental manipulation regardless of target expression. Furthermore, in the facial emotion identification task, we also observed a main effect of the RT to detect angry faces ($B = 0.33$, s.e. = 0.15, $p = 0.036$) where slower detection of angry faces in the facial emotion identification task predicted slower RT in the experimental manipulation regardless of target expression. The full table of the mixed models can be found in the electronic supplementary material, tables S9–S15.

## 3.6. Computational modelling

### 3.6.1. Reinforcement learning model

The winning model assumed that participants learned to either copy or not copy the expression of each target interactant. Participants updated the value of the action options using prediction errors based on the reinforcement received on that trial. In addition, the model assumed that the value of copying was

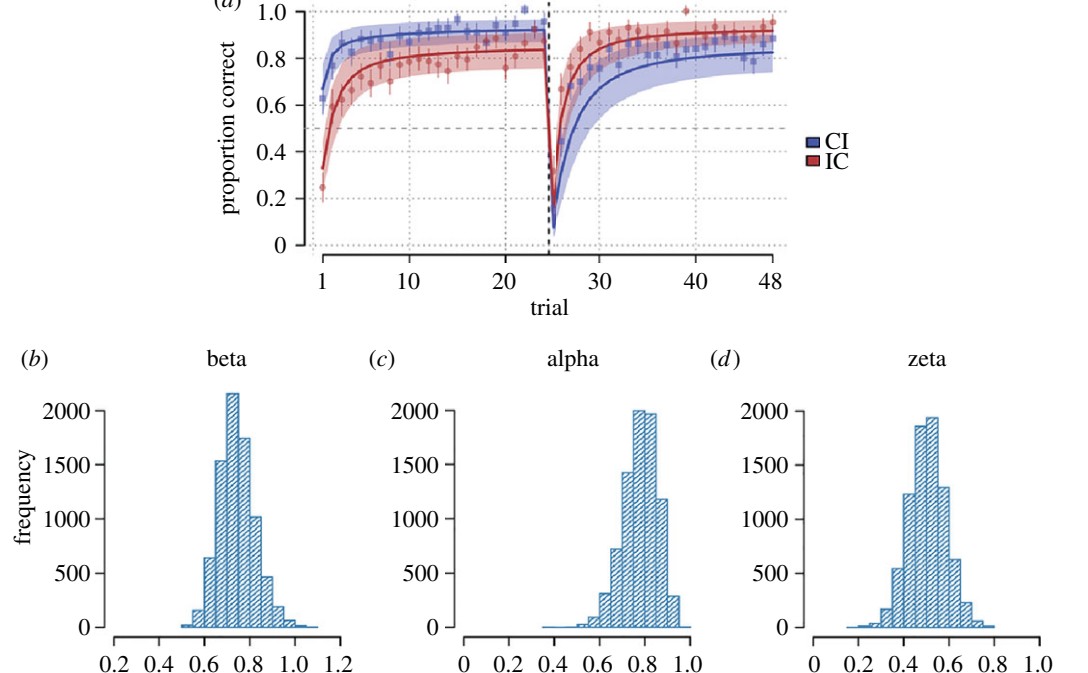

**Figure 8.** (*a*) Posterior predictions from reinforcement learning model using the full posterior (8000 samples) from each participant's parameter fits each simulated 100 times. Thick line shows the predictions average over participants and shaded region the average of each participant's 80% predictive interval. Points represent average participant data and error bars represent standard errors. Blue colour indicates trials where participants faced the target interactant to whom congruent responses were required in the first half of the experiment and incongruent responses in the second half (CI, congruent to incongruent), and red colour indicates the target interactant to which the opposite response pattern was appropriate (IC, incongruent to congruent). (*b*–*d*) Histograms showing posterior distributions for the estimated population average temperature ($\beta$), learning rate ($\alpha$) and copy-bias ($\zeta$).

affected by a static bias term which could be positive or negative. This model decisively outcompeted an alternative model without the bias term and a model that assumed participants only began the experiment with a prior bias for (or against) copying that later learning could modify. Model comparison was inconclusive when comparing against models, which allowed either learning rates or the copy bias term to vary according to what expression the target interactant was making leading us to prefer the model without those terms for reasons of parsimony. See electronic supplementary material, table 21 for full model comparison.

Figure 8 displays posterior predictions of the model and the posteriors for the estimated population average values of the three model parameters (electronic supplementary material, figures S3–S5 for per participant posteriors). The model can account for key features of the empirical data, including above/below chance performance towards the congruent/incongruent target interactant, respectively, on the first trial, as well as participants' response to the contingency reversal half-way through the experiment. Participants were on average characterized by positive $\zeta$ parameter ($M = 0.50$, 95% CrI = [0.35, 0.66], meaning an average tendency to copy the target interactants' expressions, although there was considerable variability between participants with some exhibiting the reverse tendency (electronic supplementary material, table S21).

Finally, we explored if participants' learning rates ($\alpha$) or copy-bias ($\zeta$) parameters correlated with any individual difference measures, using the maximum *a posteriori* estimate for each participant. We found a significant negative correlation between participants' STAI-T scores and the magnitude of their learning rates ($r = -28$, 95% CI = [−0.51, −0.01], $p = 0.04$); see also electronic supplementary material, figure S2. Remaining correlations were non-significant.

### 3.6.2. Drift diffusion model

To understand how participants made decisions to smile or frown, we used a drift diffusion model which takes into account both choices and RT. We used a response-coded model, where the upper boundary responses were defined as 'smile' and the lower boundary responses were defined as 'frown'. We

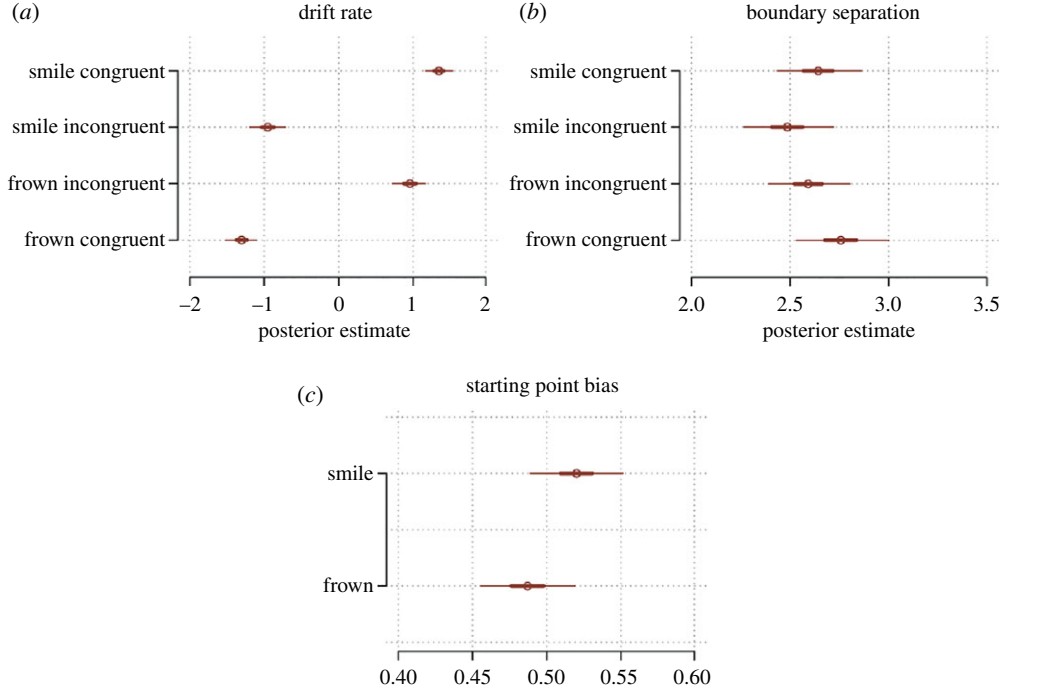

**Figure 9.** Estimated parameters of the drift diffusion model. (*a*) Drift rate and (*b*) boundary separation parameters for the four conditions in the Expression × Congruency design matrix where expression refers to the facial expression of the target facial stimuli at a given trial. (*c*) Bias, i.e. starting point of evidence accumulation as a function of target expression.

allowed the drift rate, bias and boundary separation to vary as a function of both the target facial expression and the CR on that trial as well as their interaction. The starting point bias was allowed to vary depending on the target expression. The model included random slopes for all terms. The results of our drift diffusion model are illustrated in figure 9.

### 3.6.3. Drift rate

In our model, the drift rate represented the average rate of evidence accumulation towards either making a frown or a smile response. Positive drift values represented net accumulation towards smiles while negative drift values represented net accumulation toward frowns as illustrated in figure 10*a*,*b*. We first considered if drift rates differed between congruent and incongruent conditions. To do this, we first took the absolute of drift rates in each condition and then subtracted the sum of the posterior distributions of the drift rates from the two congruent conditions from the sum of the posterior distributions from the two incongruent conditions.[1] The resulting posterior difference is plotted in figure 10*a* and shows that greater than 99.9% of the posterior mass was positive, with an average difference in drift rates estimated to be 0.77, 95% CrI = [0.30, 1.24]. This means that the magnitude of drift was on average larger in congruent trials compared with incongruent trials, suggesting that participants integrated more evidence towards a response per unit time in this condition compared with the incongruent condition. Next, we conducted the analogous comparison of target expression while marginalizing over the congruency condition. The resulting posterior difference (target smile − target frown) is depicted in figure 10*b*. The posterior mass was clearly centred over zero (*M* = 0.07, 95% CrI = [−0.14, 0.28]. In other words, average magnitude of drift rate did not differ when participants faced a frowning or smiling target interactant.

### 3.6.4. Boundary separation

We performed the same analyses on the boundary separation parameter, which represents the quantity of evidence accumulated prior to giving a response, and hence can be thought of governing response caution.

[1]Drift rates in Smile Incongruent and Frown Incongruent conditions were multiplied by −1 to make all estimates directionally consistent and comparable.

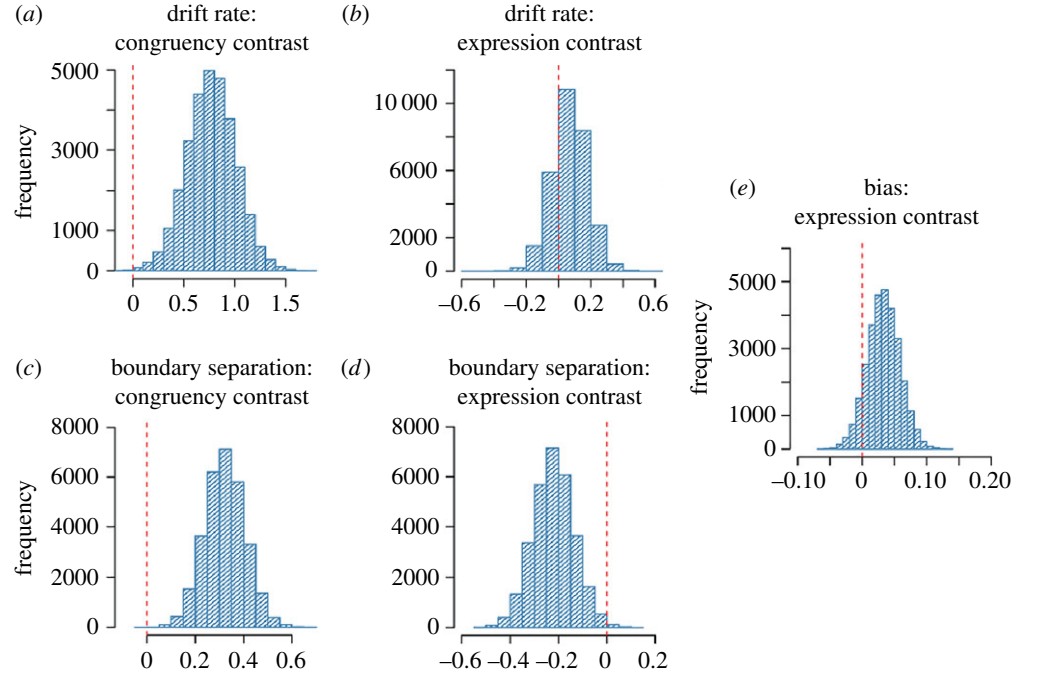

**Figure 10.** Histograms over posterior contrasts for parameters of the drift diffusion model. (*a*) Contrast between congruency conditions, marginalizing over target expression, on the drift rate. (*b*) Contrast between target expression on drift rate, marginalizing over congruency conditions. (*c*) Contrast between congruency conditions, marginalizing over target expression, on boundary separation. (*d*) Contrast between target expression on boundary separation, marginalizing over congruency conditions. (*e*) Contrast between target expression on starting point bias parameter.

Higher boundary separation values tend to mean higher-quality choices (i.e. higher likelihood of making an upper boundary choice for a positive drift rate and lower boundary choice for a negative drift rate).

Average boundary separation was higher in the congruent condition compared with the incongruent condition (figure 10*c*) with an average difference of 0.32, 95% CrI = [0.16, 0.49] and greater than 99.9% of posterior mass being positive. This meant that participants were less likely to commit speed-accuracy trade-off errors on congruent trials versus incongruent trials. Comparing boundary separation by target expression instead showed that boundary separation was on average lower when the target interactant smiled compared with frowned (M = −0.22, 95% CrI = [−0.38, −0.06]) with 99.6% of posterior mass being negative (figure 10*d*).

### 3.6.5. Starting point bias

The starting point bias quantifies if there is a tendency for the evidence accumulation to begin favouring one response or the other, measured as a proportion of the distance between the upper and lower boundaries (0.5 being the neutral middle). Contrasting the bias parameter based on the target interactants' expression we found a probable, but not certain, tendency for bias being higher (favouring smile response) when participants faced a smiling target interactant compared with when they faced a frowning target interactant (M = 0.03, 95% CrI = [−0.02, 0.08]) with 90.7% of posterior samples being positive (figure 10*e*).

## 4. Discussion

Facial expressions are the dominant non-verbal means of communication in the human species, and the ability to flexibly learn to adjust one's own facial expressions in social situations is critical for thriving in our social world. Here, we aimed to create a simplified experimental model of such social situations. Using a novel method, we examined how participants learn to adjust their facial expressions when facing a target interactant to avoid aversive outcomes. We draw five main conclusions from our results.

First, our method was successful in modelling interactive facial behaviour in a controlled experimental setting using online biofeedback based on EMG signals. We found that participants'

probability of making a CR increased across trials, showing that the participants learned the experimental contingencies and were able to improve their decision-making over time. The validation of our novel method opens up for several new research questions related to the real-time exchange of spontaneous facial expressions under controlled settings where there is a goal to avoid an aversive outcome.

Second, as hypothesized, we observed a clear effect of congruency, showing that participants exhibited improved performance on congruent compared with incongruent trials indicating that congruent exchange of facial expressions optimizes facial expression selection, thus decreasing punishment. Using drift diffusion models to capture the information processing underlying the selection and timing of facial responses, we showed that congruency affected both the drift and boundary separation parameters of the model. The greater boundary separation for the congruent versus incongruent conditions indicated that participants were less likely to engage in speed–accuracy trade-off errors in congruent trials as a consequence of engaging slower responses in these trials. However, the greater drift rates for congruent trials resulted ultimately in both faster average RT and CR as we observed in the congruent conditions.

Third, as expected, we observed that participants performed better on trials where the target interactant smiled compared with when they frowned. Interestingly, this main effect of expression was enhanced whenever the participants had to congruently respond to a smiling target stimulus versus a frowning target stimulus as revealed by our mixed effect models. These results are not surprising since reciprocating frowns are more likely to signal danger than reciprocating smiles in most situations [58–60]. Our drift diffusion model results provided a possible explanation for this: showing that there was a bias for participants to reciprocate smiling versus frowning target interactants. Our patterns of findings provide novel insights into social-contextual views of emotional mimicry [58–60], by demonstrating that the increased mimicry toward smiles versus frowns is not caused by differences in the rate of evidence accumulation toward choosing to form either expression (drift rate), but rather because less evidence is required to reciprocate a smile versus a frown (bias). Our findings and methods will aid theorists developing models of facial mimicry and the use of DDM decomposition will probably be of considerable further use when paired with other neural and physiological correlates with the aim to provide an understanding of the mechanisms underlying the decision-making of facial information exchange.

To further extend our DDM analyses approach, we provided a complementary model-based analysis, using RL models. These results demonstrated that (i) the focus of learning was learning to copy (or not) the target interactants' expressions rather than learning specific target interactant-expression contingencies; (ii) learning was best characterized by a model that included a fixed bias to copy the target interactants' expression instead of models that assumed participants only began the experiment with a prior bias for (or against) copying that later learning could modify; and finally (iii) learning was not affected by whether the target interactant smiled or frowned at them. Together, this suggests that a simple learning algorithm, which has been widely applied to a wide range of tasks in social neuroscience [10,36] can account for participants' adaptive behaviour.

Fourth, participants showed higher CR on trials on congruent versus incongruent trials after reversal as compared with before reversal. A possible explanation for the greater CR on congruent trials may be that participants needed additional trials to fully relearn the contingencies after the reversal. In order to examine this conjecture, future studies should increase the number of trials in the experiment to make performance converge at baseline. If the increased trial number does not alter the results, similar findings might capture a bias towards pre-existing social impressions [61]. However, since the reversal variable was added on an explorative basis, our interpretation of the results remains somewhat speculative.

Fifth, we tested whether individual differences in N-back task, facial emotion identification test, STAI and ERQ could explain differences in performance on the interactive task. The N-back task and facial emotion identification task did not predict CR during the experimental manipulation, implying that the effects in the main task could not be explained by individual differences in working memory and emotion recognition. We noted that increased STAI-T predicted higher CR as revealed by our mixed effects models while our selected RL model predicted lower learning rates as STAI-T increased. The results of these two different models contradict each other and are very difficult to interpret, hence, future studies should use larger sample sizes in order to test whether these contradictory results are replicated. In summary, since our analyses of individual difference measures might be underpowered, we are hesitant to draw any firm conclusion. Future studies using robust power analyses with sufficient sample sizes could explore the influence of these different individual measurements during facial information exchange.

# 5. Summary, limitations and future directions

In sum, we introduced a novel online biofeedback method to simulate avoidance learning in facial expression exchange, in which facial expressions of unknown target interactants had real and aversive consequences for the individual. We showed that individuals learn to optimize their facial expression selection over time in experimentally simulated interactive dyads in order to avoid aversive feedback, and that congruency of facial expression exchange strongly facilitates this process. Furthermore, we applied drift diffusion and reinforcement models to provide a mechanistic explanation for our findings which helped clarifying the underlying learning and decision-making processes of our study. In the present study, we limited our investigation to the exchange of two transient emotional expressions: smiles and frowns. The reality of social interaction is no doubt far more complex than our paradigm affords, and our experimental paradigm therefore suffers from a limited generalizability. For example, our paradigm was designed as a two-alternative forced choice (2AFC) where the aim was to introduce a novel method of studying facial expression decisions. However, it remains to be determined whether our method can be generalized to multiple-alternative forced choice (mAFC) tasks [62] or tasks where participants can freely choose which facial expression to form (e.g. an unconstrained trial and error design). It is also an open question, highlighted by the copy-bias RL model that we found best explained trial-by-trial learning, if people make choices to copy or not copy in m-alternative settings or if they select directly from a repertoire of available facial expressions. Addressing such questions will be easier using methods like the one introduced in this paper.

Our aim was to provide an experimental model of real-life interactive exchange of facial information involving feedback, hoping that this would open up new avenues to study increasingly complex aspects of social interactions. In real life, our interaction partners also differ across various socially relevant dimensions such as facial features. Hence, future studies should examine if socially relevant traits, such as gender, age, dominance and trustworthiness [63] affect the interactive decision-making process involved in the exchange of emotional expressions. In the present study, we used pre-recorded videos of different northern European males and females. Future studies should use psychophysics-driven approaches to parametrize anatomical features, and include a more ethnically diverse set of male and female faces, and furthermore, we should collect information on participants' ethnic backgrounds. Another important topic for future research is to examine how different motivational goals, such as cooperation and competition, influence facial expression selection bias in interactive situations [64]. Furthermore, our study did to not confirm whether the learning processes in this study were specific to facial expression recognition and mimicry or domain general in nature. Future studies could, for example, use our paradigm to investigate learning to respond to social (e.g. arbitrary mouth shapes) and non-social stimuli, such as written prompts to smile and frown. Finally, future studies should collect structured information about intentions ascribed to the target interactants, participants' learning strategies.

We hope that the results presented here will contribute to the important pursuit of increasing our knowledge of learning and decision-making processes in real-life, interactive, settings, which poses an important challenge facing psychology in the coming decades [65]. Through translational efforts, our findings might also hold a promise for a better understanding of the bases of many psychological dysfunctions characterized by impaired learning and decision-making in social situations.

Ethics. All participants were recruited at the Karolinska Institutet (Stockholm, Sweden) campus and provided written consent. All participants provided a written and informed consent prior to participation. The present study was approved by the Regional Ethic Committee of Stockholm (Dnr: 2017/37-31/4). Participants received two cinema vouchers as compensation.

Data accessibility. The data and scripts presented and used in this article can be found at the Open Science Framework: https://osf.io/f7edp/?view_only=ddf061b110e2479f972e069269e02b26.

Authors' contributions. Author contributions were as follows: all authors contributed to the study concept. All authors contributed to the study design. Method development, testing and data collection were performed by J.Y. All authors performed the data analysis. All authors drafted the manuscript, A.O. and P.P. provided critical revisions. A.O. and P.P. jointly supervised the project. All authors approved the final version of the manuscript for submission. Results reported in this article were partially presented at the 11th Annual Meeting of the Social & Affective Neuroscience Society in, 2018 in Brooklyn, NY and at the 12th Annual Meeting of the Social & Affective Neuroscience Society in 2019 in Miami as part of the conference proceedings.

Competing interests. We declare we have no competing interests.

Funding. A.O. was supported by the Knut and Alice Wallenberg Foundation (KAW 2014.0237), a European Research Council Starting grant (no. 284366; Emotional Learning in Social Interaction project), and a Consolidator grant (no. 2018-00877) from Swedish Research Foundation (VR).

Acknowledegements. We thank Fredrik Rådebjörk at JoR AB for assisting us in developing the method in this study.

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
