## [Peer Review File · Royal Society Open Science]

Review History

RSOS-202159.R0 (Original submission)

Review form: Reviewer 1

Is the manuscript scientifically sound in its present form?

Yes

Are the interpretations and conclusions justified by the results?

Yes

Is the language acceptable?

Yes

Do you have any ethical concerns with this paper?

No

Have you any concerns about statistical analyses in this paper?

No

Recommendation?

Major revision is needed (please make suggestions in comments)

Comments to the Author(s)

I found myself being of two minds about this paper. On the one hand, it was well-grounded, technically proficient, and very inventive. On the other hand, I was left a little uncertain as to the overall takeaway message. Overall, participants learned to copy or not copy targets' expressions accordingly, and this learning was better when targets required congruent expressions and when targets smiled. This is very interesting, but does this pattern of data reflect general principles guiding social (or non-social) reinforcement learning, or something specific to facial expression recognition and mimicry? For example, would the same results have been observed if targets were making (and participants had to mimic) something other than emotional expressions (for example, mouth shapes representing phonemes)? Would the same results have been observed if the target stimuli were just the words "smile" and "frown"? The grounding of this manuscript is about the necessity of modelling our facial vocabulary off of the displays of others (which would seem to require higher-level mental-state inference and attribution; see below), but the learning on display in this work seems like it could (and can, given the modeling data) be explained by relatively simple reinforcement.

Is the kind of mimicry that the authors are interested (e.g., as it unfolds in vivo) really just a pure stimulus-response association? For example, the conflict posed in the initial example ("You may take a confrontational approach and reciprocate the angry expression, thus risking conflict, or take a more submissive approach by smiling, and thereby increase the chances of avoiding harm") reflects underlying inferences about this agent... their expression reflects an emotional state of anger, and therefore, reciprocation or non-reciprocation reflects an understanding of and response to that emotional state.

This led me to wonder – do the authors think that participants are forming impressions of these social targets and does impression formation aid (or impede) reward contingency learning in the task? For example, rather than just learning to copy this face and do the opposite of that face, one prediction might be that people infer that the former target is helpful [e.g., he's making it easier on me] and the other is a hindrance [e.g., he's trying to trick me] – perhaps to the extent that participants make these *trait* inferences, their reward learning is improved. Did the authors collect any data on participants' evaluations of these targets? Similarly, if the targets' expressions are inferred to reflect their internal emotional states, is there any evidence that participants made such intuitions? Lastly, was there any sort of structured debriefing to get an understanding of how participants made sense of the task? (e.g., Did they report forming some sort of rule or strategy to govern their responses? Did they think of the shocks as coming from these targets or independent from them?)

2. With regards to the modeling, it seemed like there was some uncertainty as to whether to make this the focal point of the manuscript, or a set of supplementary analyses. (Indeed, in the abstract, the authors note that these approaches can reveal key aspects of the mechanisms guiding the learning herein, but then don't elaborate there.) I wonder if it might be possible to foreground the specifics of their modeling-related predictions in the text – personally, I felt like I didn't have a full grasp on how these data should be integrated into the overall picture until the Discussion. (I recognize that the authors do something like this on pages 7-8, so maybe this is just me... even so, some portions of this section were a bit imprecise – e.g., the authors write, "We evaluated several competing models of participants' learning process," but don't yet give the reader a sense of what different models might capture this learning.

3. This is a somewhat more minor point – I also wondered a bit about how well the incongruent condition actually fits with the overarching framing. The authors describe the problem at hand

like so: "For example, not realizing that an interlocutor's smile has changed from signaling friendliness to cold politeness when engaging in casual banter can lead to social gaffes or damaged relationships." That's very true and presents an interesting question! From an ecological validity standpoint, this isn't what the incongruent condition does though. The right analogue would be a person who smiles when they're happy and frowns when they're sad.

4. A bit more information regarding the stimuli would be helpful. The authors write "Twelve video clips were retrieved from the Amsterdam Dynamic Facial Expression Set (ADFES) (van der Schalk et al., 2011)" on page 10. That being said, what were the selection criteria? Were these stimuli normed in terms of expression prototypicality and intensity? Were they equated across stimuli within gender? Were they equated between stimuli *across* gender? Given the differences between the smiling and frowning expressions (and the integration of participants' EMG responses into the feedback), it would of course be an issue if the frowns were less intense or less prototypic than the smiles.

Also, I wasn't totally sure I had the following details right... the authors pulled three target identities within gender (e.g., 3 male, 3 female) in total from the larger set, but each individual participant only saw *two* targets (each making two different expressions at different points during the experiment). Is that accurate? If so, how were the stimulus selections managed at the individual subject level?

5. The authors gender-matched targets to participants and they had a good gender balance of participants. Given these details, did they assess whether participant gender moderated any of the learning effects? Moreover, since all the targets were white, did they also restrict participant recruitment based on race? (If so, please note participant race in the Methods. If not, why not? Participant and target race have a demonstrable impact on emotion recognition and mimicry.)

6. There are number of sections in the manuscript that could really use a second read-through and careful revision. For example, there's some important detail in the following excerpt from the Methods, but some of it is very hard to parse:

"Six of the video clips consisted of three different male faces and the male faces where gender matched to male participants. Each male face had a subset of two video clips whereas is one of them, he expressed happiness and in the other video clip, expressed anger upon presentation."

Here's another one right in the opening on page 3: "Therefore, an important question is to determine what the mechanisms underlying the learning of, and deciding about, our facial behaviours during interactions in a threatening environment are?"

Finally, there are a fair number of subject/verb agreement issues (especially in the Intro; e.g., "spontaneous facial mimicry facilitate congruent exchanges of facial expressions," "such as when an individual learn to frown," etc.) that should be revised.

7. Lastly, on page 31, the authors write, "Additionally, we observed that higher ESF (Expression sub-facet of ERQ) predicted higher CR, suggesting that suppressing one's own emotional expression could facilitate a better performance." I could be missing something, but I couldn't find this detail in the main text; it seems like it would be in the section on page 22 but isn't. (At the same time, in that section, they refer to an effect of STAI but the associated stats don't seem to represent a statistically significant result: "Finally, we observed a main effect of STAI-T ($B = 0.75$, $SE = 0.14$, $p = 0.34$) where increased STAI-T predicted lower CR." I'm not sure what's going on here, though I may be missing something.)

Review form: Reviewer 2

Is the manuscript scientifically sound in its present form?

Yes

Are the interpretations and conclusions justified by the results?

Yes

Is the language acceptable?

Yes

Do you have any ethical concerns with this paper?

No

Have you any concerns about statistical analyses in this paper?

No

Recommendation?

Accept with minor revision (please list in comments)

Comments to the Author(s)

The manuscript presents a novel method to examine how humans learn to produce adaptive facial responses to the facial expressions of others. Participants viewed videos of smiling or frowning faces and learned which facial response (either congruent or incongruent) are associated with avoidance of electrical shock. Results showed quicker learning with congruent facial responses, particularly smiling. The manuscript concludes that the presented method could be used to further knowledge social interaction processes.

The manuscript presents a promising method and initial results that demonstrate its potential to advance understanding of complex social interaction processes. The manuscript is very well written, and the experimental work is thorough and well considered. Overall, the manuscript presents a convincing demonstration of the method and highlights well the new research avenues that could be explored with this method. To better support this aim, certain aspects of the manuscript could be strengthened, as outlined below.

The method is designed primarily for the study of human social interactions, which, as the manuscript describes well, is complex. However, the task used here is maximally simplified – i.e., a 2AFC A-not-A facial response task. A first issue is whether this simplified task is too easy, particularly in a trial-and-error situation. For example, on the very first trial, the participant only needs to choose one facial response at random to learn what the correct response is. A second issue is whether this 2AFC-based method, including the computational modelling components, can be generalized to higher N-AFC tasks (e.g., see DeCarlo, 2012, *Journal of Mathematical Psychology*) or where participants can freely choose which facial response to make (i.e., an unconstrained trial and error design).

The main aim of the task is to measure the facial responses of participants in response to videos of others' facial expressions. However, it's not clear whether facial responses were recorded during the videos or for how long facial responses were recorded. For example, a small involuntary facial movement could be made initially followed by the facial movement the participant intends to make. Which would be considered for recording of correct vs incorrect trials? A related question is whether certain facial muscles are more or less under voluntary control and whether the speed of activation differs.

Some discussion points could be developed. Specifically, learning is quicker with congruent facial expressions, particularly for smiling. Might this reflect frequency of such responses? For example, most interactions are friendly with smiling being met with smiles in return and threat-related facial expressions being rarer.

To strengthen claims that this method could be used to better understand social interactions, it would be useful to report the pattern of results per participant – that is, does the pattern of results reported replicate across participants (vs reflects an averaging artefact, see also Grice et al., 2020 *Advances in Methods and Practices in Psychological Science*)?

Minor points:

- Report the visual angle, in degrees (horizontal and vertical), of the face stimuli in each experiment
- Explain why the face stimuli were gender matched to participants?
- Provide a rationale for the threshold of 0.001mV/s
- Mean RTs are computed though it's not clear that the underlying data meet the assumptions of parametric analyses
- By "Caucasian" do the authors mean "white"?
- Given that the ethnicity of stimuli and participant could impact the results, report both. Similarly, report the cultural background of participants
- The section Materials, Stimuli could be more clearly written/expressed
- "Males" and "females" or "men" and "women"
- Page 11, line 29/30 "clips of a face" (not "on a face")
- "Our" species could be better described as "the human" species
- See Zhan et al., (2019) *NHB* for parametric control of face features

Decision letter (RSOS-202159.R0)

Dear Mr Yi

The Editors assigned to your paper RSOS-202159 "The face value of feedback: Facial behaviour is shaped by goals and punishments during interaction with dynamic faces" have now received comments from reviewers and would like you to revise the paper in accordance with the reviewer comments and any comments from the Editors. Please note this decision does not guarantee eventual acceptance.

Please submit your revised manuscript and required files (see below) no later than 21 days from today's (ie 26-Jan-2021) date. Note: the ScholarOne system will 'lock' if submission of the revision is attempted 21 or more days after the deadline. If you do not think you will be able to meet this deadline please contact the editorial office immediately.

on behalf of Dr Giorgia Silani (Associate Editor) and Essi Viding (Subject Editor)
openscience@royalsociety.org

Associate Editor Comments to Author (Dr Giorgia Silani):

While the reviewers found your work to be potentially important and conceptually appropriate for RSOS, they have highlighted some weakness and provided constructive suggestions that would need to be addressed before the manuscript would be considered for publication. Thus, I would be glad to reconsider a revised manuscript which takes into account the points raised by the reviewers.

Reviewer comments to Author:

Reviewer: 1

Comments to the Author(s)

I found myself being of two minds about this paper. On the one hand, it was well-grounded, technically proficient, and very inventive. On the other hand, I was left a little uncertain as to the overall takeaway message. Overall, participants learned to copy or not copy targets' expressions accordingly, and this learning was better when targets required congruent expressions and when targets smiled. This is very interesting, but does this pattern of data reflect general principles guiding social (or non-social) reinforcement learning, or something specific to facial expression recognition and mimicry? For example, would the same results have been observed if targets were making (and participants had to mimic) something other than emotional expressions (for example, mouth shapes representing phonemes)? Would the same results have been observed if the target stimuli were just the words "smile" and "frown"? The grounding of this manuscript is about the necessity of modelling our facial vocabulary off of the displays of others (which would seem to require higher-level mental-state inference and attribution; see below), but the learning on display in this work seems like it could (and can, given the modeling data) be explained by relatively simple reinforcement.

Is the kind of mimicry that the authors are interested (e.g., as it unfolds in vivo) really just a pure stimulus-response association? For example, the conflict posed in the initial example ("You may

take a confrontational approach and reciprocate the angry expression, thus risking conflict, or take a more submissive approach by smiling, and thereby increase the chances of avoiding harm”) reflects underlying inferences about this agent... their expression reflects an emotional state of anger, and therefore, reciprocation or non-reciprocation reflects an understanding of and response to that emotional state.

This led me to wonder – do the authors think that participants are forming impressions of these social targets and does impression formation aid (or impede) reward contingency learning in the task? For example, rather than just learning to copy this face and do the opposite of that face, one prediction might be that people infer that the former target is helpful [e.g., he’s making it easier on me] and the other is a hindrance [e.g., he’s trying to trick me] – perhaps to the extent that participants make these *trait* inferences, their reward learning is improved. Did the authors collect any data on participants’ evaluations of these targets? Similarly, if the targets’ expressions are inferred to reflect their internal emotional states, is there any evidence that participants made such intuitions? Lastly, was there any sort of structured debriefing to get an understanding of how participants made sense of the task? (e.g., Did they report forming some sort of rule or strategy to govern their responses? Did they think of the shocks as coming from these targets or independent from them?)

2. With regards to the modeling, it seemed like there was some uncertainty as to whether to make this the focal point of the manuscript, or a set of supplementary analyses. (Indeed, in the abstract, the authors note that these approaches can reveal key aspects of the mechanisms guiding the learning herein, but then don’t elaborate there.) I wonder if it might be possible to foreground the specifics of their modeling-related predictions in the text – personally, I felt like I didn’t have a full grasp on how these data should be integrated into the overall picture until the Discussion. (I recognize that the authors do something like this on pages 7-8, so maybe this is just me... even so, some portions of this section were a bit imprecise – e.g., the authors write, “We evaluated several competing models of participants’ learning process,” but don’t yet give the reader a sense of what different models might capture this learning.

3. This is a somewhat more minor point – I also wondered a bit about how well the incongruent condition actually fits with the overarching framing. The authors describe the problem at hand like so: “For example, not realizing that an interlocutor’s smile has changed from signaling friendliness to cold politeness when engaging in casual banter can lead to social gaffes or damaged relationships.” That’s very true and presents an interesting question! From an ecological validity standpoint, this isn’t what the incongruent condition does though. The right analogue would be a person who smiles when they’re happy and frowns when they’re sad.

4. A bit more information regarding the stimuli would be helpful. The authors write “Twelve video clips were retrieved from the Amsterdam Dynamic Facial Expression Set (ADFES) (van der Schalk et al., 2011)” on page 10. That being said, what were the selection criteria? Were these stimuli normed in terms of expression prototypicality and intensity? Were they equated across stimuli within gender? Were they equated between stimuli *across* gender? Given the differences between the smiling and frowning expressions (and the integration of participants’ EMG responses into the feedback), it would of course be an issue if the frowns were less intense or less prototypic than the smiles.

Also, I wasn’t totally sure I had the following details right... the authors pulled three target identities within gender (e.g., 3 male, 3 female) in total from the larger set, but each individual participant only saw *two* targets (each making two different expressions at different points during the experiment). Is that accurate? If so, how were the stimulus selections managed at the individual subject level?

5. The authors gender-matched targets to participants and they had a good gender balance of participants. Given these details, did they assess whether participant gender moderated any of the learning effects? Moreover, since all the targets were white, did they also restrict participant recruitment based on race? (If so, please note participant race in the Methods. If not, why not? Participant and target race have a demonstrable impact on emotion recognition and mimicry.)

6. There are number of sections in the manuscript that could really use a second read-through and careful revision. For example, there's some important detail in the following excerpt from the Methods, but some of it is very hard to parse:

"Six of the video clips consisted of three different male faces and the male faces where gender matched to male participants. Each male face had a subset of two video clips whereas is one of them, he expressed happiness and in the other video clip, expressed anger upon presentation."

Here's another one right in the opening on page 3: "Therefore, an important question is to determine what the mechanisms underlying the learning of, and deciding about, our facial behaviours during interactions in a threatening environment are?"

Finally, there are a fair number of subject/verb agreement issues (especially in the Intro; e.g., "spontaneous facial mimicry facilitate congruent exchanges of facial expressions," "such as when an individual learn to frown," etc.) that should be revised.

7. Lastly, on page 31, the authors write, "Additionally, we observed that higher ESF (Expression sub-facet of ERQ) predicted higher CR, suggesting that suppressing one's own emotional expression could facilitate a better performance." I could be missing something, but I couldn't find this detail in the main text; it seems like it would be in the section on page 22 but isn't. (At the same time, in that section, they refer to an effect of STAI but the associated stats don't seem to represent a statistically significant result: "Finally, we observed a main effect of STAI-T ($B = 0.75$, $SE = 0.14$, $p = 0.34$) where increased STAI-T predicted lower CR." I'm not sure what's going on here, though I may be missing something.)

Reviewer: 2

Comments to the Author(s)

The manuscript presents a novel method to examine how humans learn to produce adaptive facial responses to the facial expressions of others. Participants viewed videos of smiling or frowning faces and learned which facial response (either congruent or incongruent) are associated with avoidance of electrical shock. Results showed quicker learning with congruent facial responses, particularly smiling. The manuscript concludes that the presented method could be used to further knowledge social interaction processes.

The manuscript presents a promising method and initial results that demonstrate its potential to advance understanding of complex social interaction processes. The manuscript is very well written, and the experimental work is thorough and well considered. Overall, the manuscript presents a convincing demonstration of the method and highlights well the new research avenues that could be explored with this method. To better support this aim, certain aspects of the manuscript could be strengthened, as outlined below.

The method is designed primarily for the study of human social interactions, which, as the manuscript describes well, is complex. However, the task used here is maximally simplified – i.e., a 2AFC A-not-A facial response task. A first issue is whether this simplified task is too easy, particularly in a trial-and-error situation. For example, on the very first trial, the participant only needs to choose one facial response at random to learn what the correct response is. A second

issue is whether this 2AFC-based method, including the computational modelling components, can be generalized to higher N-AFC tasks (e.g., see DeCarlo, 2012, *Journal of Mathematical Psychology*) or where participants can freely choose which facial response to make (i.e., an unconstrained trial and error design).

The main aim of the task is to measure the facial responses of participants in response to videos of others' facial expressions. However, it's not clear whether facial responses were recorded during the videos or for how long facial responses were recorded. For example, a small involuntary facial movement could be made initially followed by the facial movement the participant intends to make. Which would be considered for recording of correct vs incorrect trials? A related question is whether certain facial muscles are more or less under voluntary control and whether the speed of activation differs.

Some discussion points could be developed. Specifically, learning is quicker with congruent facial expressions, particularly for smiling. Might this reflect frequency of such responses? For example, most interactions are friendly with smiling being met with smiles in return and threat-related facial expressions being rarer.

To strengthen claims that this method could be used to better understand social interactions, it would be useful to report the pattern of results per participant – that is, does the pattern of results reported replicate across participants (vs reflects an averaging artefact, see also Grice et al., 2020 *Advances in Methods and Practices in Psychological Science*)?

Minor points:

- Report the visual angle, in degrees (horizontal and vertical), of the face stimuli in each experiment
- Explain why the face stimuli were gender matched to participants?
- Provide a rationale for the threshold of 0.001mV/s
- Mean RTs are computed though it's not clear that the underlying data meet the assumptions of parametric analyses
- By "Caucasian" do the authors mean "white"?
- Given that the ethnicity of stimuli and participant could impact the results, report both. Similarly, report the cultural background of participants
- The section Materials, Stimuli could be more clearly written/expressed
- "Males" and "females" or "men" and "women"
- Page 11, line 29/30 "clips of a face" (not "on a face")
- "Our" species could be better described as "the human" species
- See Zhan et al., (2019) *NHB* for parametric control of face features

===PREPARING YOUR MANUSCRIPT===

Please ensure that you include an acknowledgements' section before your reference list/bibliography. This should acknowledge anyone who assisted with your work, but does not

qualify as an author per the guidelines at <https://royalsociety.org/journals/ethics-policies/openness/>.

===PREPARING YOUR REVISION IN SCHOLARONE===

- Ensure that your data access statement meets the requirements at <https://royalsociety.org/journals/authors/author-guidelines/#data>. You should ensure that you cite the dataset in your reference list. If you have deposited data etc in the Dryad repository, please include both the 'For publication' link and 'For review' link at this stage.
- If you are requesting an article processing charge waiver, you must select the relevant waiver option (if requesting a discretionary waiver, the form should have been uploaded at Step 3 'File upload' above).
- If you have uploaded ESM files, please ensure you follow the guidance at <https://royalsociety.org/journals/authors/author-guidelines/#supplementary-material> to include a suitable title and informative caption. An example of appropriate titling and captioning may be found at https://figshare.com/articles/Table_S2_from_Is_there_a_trade-off_between_peak_performance_and_performance_breadth_across_temperatures_for_aerobic_scope_in_teleost_fishes_/3843624.

Author's Response to Decision Letter for (RSOS-202159.R0)

See Appendix A.

RSOS-202159.R1 (Revision)

Review form: Reviewer 1

Is the manuscript scientifically sound in its present form?

Yes

Are the interpretations and conclusions justified by the results?

Yes

Is the language acceptable?

Yes

Do you have any ethical concerns with this paper?

No

Have you any concerns about statistical analyses in this paper?

No

Recommendation?

Accept as is

Comments to the Author(s)

I thank the authors for their careful attention and thoughtful replies to my review. I really appreciated their thoroughness in responding to my concerns and editing the manuscript accordingly. While my initial appraisal of the manuscript was already quite positive, I think it is improved even further!

I do still have *one* lingering concern regarding the need for additional detail regarding the stimuli. As I mentioned in my first review, ideally, targets' smiles would be as intense and as recognizable as their frowns. While the authors didn't provide any additional norming data on these stimuli, they did demonstrate that their effects maintained when controlling for stimulus identity. That said, I'll just note that the authors of the ADFES *did* collect valence, arousal, and emotion recognition data for these stimuli (e.g., pgs. 910-911 in van der Schalk et al., 2011) so presumably this information is available. If it's at all possible to obtain these details for purposes of characterizing the stimuli, I think this would be useful to the reader.

With regards to emotion recognition, I will admit that I'm now a little confused as to why the authors chose to link frown expressions to anger (rather than sadness). (The experiment instructions specifically directed participants to frown, but the authors used anger expressions from the ADFES.) While they state in their rebuttal that "all the facial stimuli within the ADFES dataset were aimed primarily to capture the "core" action units (AUs) for the respective expressions (smile and frown) within the FACS system," the ADFES paper suggested that frowning was not reflected in the AUs of their anger expressions. "Lips tightened/pressed" (e.g., AUs 23 & 24) appears for anger, but "lip corner depress" does not (AU15 *does* appear for sadness). Ultimately, for me, this raised a possible concern would be that it was easier for participants to associate smiles with happy expressions than it was to associate frowns with anger expressions.

Decision letter (RSOS-202159.R1)

Dear Mr Yi,

It is a pleasure to accept your manuscript entitled "The face value of feedback: Facial behaviour is shaped by goals and punishments during interaction with dynamic faces" in its current form for publication in Royal Society Open Science. The comments of the reviewer(s) who reviewed your manuscript are included at the foot of this letter.

If you have not already done so, please ensure that the data deposition associated with this work is made public as soon as possible.

on behalf of Dr Giorgia Silani (Associate Editor) and Essi Viding (Subject Editor)
openscience@royalsociety.org

Associate Editor Comments to Author (Dr Giorgia Silani):

Associate Editor: 1

Comments to the Author:

I am happy to inform you that all reviewers agreed on considering the paper ready for publication. Few suggestions are still pending from reviewer 1. I kindly ask you to address them, before proceeding.

Reviewer comments to Author:

Reviewer: 1

Comments to the Author(s)

I thank the authors for their careful attention and thoughtful replies to my review. I really appreciated their thoroughness in responding to my concerns and editing the manuscript accordingly. While my initial appraisal of the manuscript was already quite positive, I think it is improved even further!

I do still have *one* lingering concern regarding the need for additional detail regarding the stimuli. As I mentioned in my first review, ideally, targets' smiles would be as intense and as recognizable as their frowns. While the authors didn't provide any additional norming data on these stimuli, they did demonstrate that their effects maintained when controlling for stimulus identity. That said, I'll just note that the authors of the ADFES *did* collect valence, arousal, and emotion recognition data for these stimuli (e.g., pgs. 910-911 in van der Schalk et al., 2011) so presumably this information is available. If it's at all possible to obtain these details for purposes of characterizing the stimuli, I think this would be useful to the reader.

With regards to emotion recognition, I will admit that I'm now a little confused as to why the authors chose to link frown expressions to anger (rather than sadness). (The experiment instructions specifically directed participants to frown, but the authors used anger expressions from the ADFES.) While they state in their rebuttal that "all the facial stimuli within the ADFES dataset were aimed primarily to capture the "core" action units (AUs) for the respective expressions (smile and frown) within the FACS system," the ADFES paper suggested that frowning was not reflected in the AUs of their anger expressions. "Lips tightened/pressed" (e.g., AUs 23 & 24) appears for anger, but "lip corner depress" does not (AU15 *does* appear for

sadness). Ultimately, for me, this raised a possible concern would be that it was easier for participants to associate smiles with happy expressions than it was to associate frowns with anger expressions.

Appendix A

Dear Mr Yi

The Editors assigned to your paper RSOS-202159 "The face value of feedback: Facial behaviour is shaped by goals and punishments during interaction with dynamic faces" have now received comments from reviewers and would like you to revise the paper in accordance with the reviewer comments and any comments from the Editors. Please note this decision does not guarantee eventual acceptance.

Please submit your revised manuscript and required files (see below) no later than 21 days from today's (ie 26-Jan-2021) date. Note: the ScholarOne system will 'lock' if submission of the revision is attempted 21 or more days after the deadline. If you do not think you will be able to meet this deadline please contact the editorial office immediately.

Best regards,

on behalf of Dr Giorgia Silani (Associate Editor) and Essi Viding (Subject Editor)
openscience@royalsociety.org

Associate Editor Comments to Author (Dr Giorgia Silani):

While the reviewers found your work to be potentially important and conceptually appropriate for RSOS, they have highlighted some weakness and provided constructive suggestions that would need to be addressed before the manuscript would be considered for publication. Thus, I would be glad to reconsider a revised manuscript which takes into account the points raised by the reviewers.

We thank the editor for the encouraging words and the possibility to respond to the issues raised by the two reviewers. Here below, we insert our responses to the reviewers' comments. According to the guidelines, we have also submitted two copies of the revised manuscript: one version where all changes have been highlighted through colouring and one "clean" version.

Reviewer comments to Author:

Reviewer: 1

Comments to the Author(s)

I found myself being of two minds about this paper. On the one hand, it was well-grounded, technically proficient, and very inventive.

We are very happy to learn that the reviewer shares our view of the many strengths of our study.

On the other hand, I was left a little uncertain as to the overall takeaway message. Overall, participants learned to copy or not copy targets' expressions accordingly, and this learning was better when targets required congruent expressions and when targets smiled. This is very interesting, but does this pattern of data reflect general principles guiding social (or non-social) reinforcement learning, or something specific to facial expression recognition and mimicry?

Thank you for pointing out this issue. First of all, we can certainly assume that general principles of reinforcement learning were involved. Indeed, our modeling indicates that a simple associative learning model can account for what information participants acquired during the task. Furthermore, our decision-making modeling taking both choices (expressions) and response times into account, as well as previous literature, leads us to believe that participants exhibit a bias to facially reciprocate smiles more frequently than frowns. For example, previous studies exploring the theory of social-contextual views of emotional mimicry have shown that spontaneous facial mimicry is more frequent and consistent toward smiling faces than frowning faces (Keltner and Haidt, 1999, Hess &

Fischer, 2013, Parkinson, 2011), an observation ascribed to the fact that reciprocating frowns can more dangerous than reciprocating smiles in most situations.

For example, would the same results have been observed if targets were making (and participants had to mimic) something other than emotional expressions (for example, mouth shapes representing phonemes)? Would the same results have been observed if the target stimuli were just the words “smile” and “frown”? The grounding of this manuscript is about the necessity of modelling our facial vocabulary off of the displays of others (which would seem to require higher-level mental-state inference and attribution; see below), but the learning on display in this work seems like it could (and can, given the modeling data) be explained by relatively simple reinforcement.

We believe that making expressions in response to stimuli like “smile” or “frown” is different compared to when responding to actual facial expressions, and even more so from suppressing smiles or frowns in a facial mimicry context. Especially since there is a large corpus of previous literature showing that we very readily reciprocate facial expressions without conscious control. This ability to mimic expressions is likely aided by highly specialized circuitry for processing facial information, such as the fusiform face area. Nevertheless, it would certainly be of interest to contrast our current paradigm as the reviewer suggests where participants respond to words instead of faces. We hypothesize that the copy bias that we observe would be attenuated with verbal stimuli, but might still be present to some degree due to semantic associations between e.g. “smile” and the motor command to smile.

In the revised manuscript we now discuss this possibility under Summary, limitations, and future directions, Page 30, Paragraph 2.

Is the kind of mimicry that the authors are interested (e.g., as it unfolds in vivo) really just a pure stimulus-response association? For example, the conflict posed in the initial example (“You may take a confrontational approach and reciprocate the angry expression, thus risking conflict, or take a more submissive approach by smiling, and thereby increase the chances of avoiding harm”) reflects underlying inferences about this agent... their expression reflects an emotional state of anger, and therefore, reciprocation or non-reciprocation reflects an understanding of and response to that emotional state.

The initial example quoted by the reviewer refers to a situation where an individual has to respond adaptively while weighing the risk of potential harm to themselves. Our paradigm represents a gross simplification of that situation, as our participants upon facing one of the interactants in our experiment, need to consider how to avoid harm by choosing an appropriate response (facial expression). We agree with the reviewer, however, the kind of everyday situations that we attempted to model (as well as illustrated in our example) likely involve both simple S-S associations, and state and trait attributions. In our experiment, we did however not manipulate or measure any such inferences. Thus, we can only speculate whether they did so or not. In fact, we did not see any reason for why not both S-S

associations and attributions might have played a role during task performance. The potential for both manipulating and measuring trait inferences in future studies is an intriguing one, and this has been highlighted under Summary, limitations, and future directions, Page 30, Paragraph 2 in the revised version of the manuscript.

This led me to wonder—do the authors think that participants are forming impressions of these social targets and does impression formation aid (or impede) reward contingency learning in the task? For example, rather than just learning to copy this face and do the opposite of that face, one prediction might be that people infer that the former target is helpful [e.g., he’s making it easier on me] and the other is a hindrance [e.g., he’s trying to trick me] – perhaps to the extent that participants make these *trait* inferences, their reward learning is improved. Did the authors collect any data on participants’ evaluations of these targets?

When it comes to impression formation of the targets, we do not have any direct evidence which would support the hypotheses that the participants ascribed helpful or hindrance related attributes to the targets since we did not collect any of the participants evaluation of the targets. However, it is likely as the reviewer suggests that the participants form various social impressions toward the target the targets during our experimental manipulation. Hence, we believe that collecting participants’ evaluation of the targets within these socially relevant dimensions could certainly be helpful in future studies.

Similarly, if the targets’ expressions are inferred to reflect their internal emotional states, is there any evidence that participants made such intuitions? Lastly, was there any sort of structured debriefing to get an understanding of how participants made sense of the task? (e.g., Did they report forming some sort of rule or strategy to govern their responses? Did they think of the shocks as coming from these targets or independent from them?)

We did not collect any structured debriefings from the participants, however we made it very clear that the target interactants were merely virtual avatars which indeed was the case. We were transparent about this contingency from the start so there was no deception in this regard. We did not collect any information as to whether the participants thought the shocks were delivered independent from the target interactants.

But for future studies, it would be of major interest to collect structured information revolving around ascribing intentions of the target interactants as well as collecting information pertaining to the participants’ learning strategy. We briefly discuss these new perspectives under Summary, limitations, and future directions, Page 30, Paragraph 2.

2. With regards to the modeling, it seemed like there was some uncertainty as to whether to make this the focal point of the manuscript, or a set of supplementary analyses. (Indeed, in the abstract, the authors note that these approaches can reveal key aspects of the mechanisms guiding the learning herein, but then don’t elaborate there.) I wonder if it might be possible to foreground the specifics of their modeling-related predictions in the text – personally, I felt like I didn’t have a full grasp on how these data should be integrated into the overall picture until the Discussion. (I

recognize that the authors do something like this on pages 7-8, so maybe this is just me... even so, some portions of this section were a bit imprecise—e.g., the authors write, “We evaluated several competing models of participants’ learning process,” but don’t yet give the reader a sense of what different models might capture this learning.

We understand that the reviewer was not fully satisfied with our treatment of the modeling results, and we hope that the role of these results has become clearer in our revised manuscript. In our study, our main goal was to establish the method for studying adaptive facial responses. Hence, while the modeling was not the main focus we nevertheless wished to explore how dominant modeling approaches could illuminate our participants’ performance. We therefore struck a balance keeping some of the description in the main text relatively brief. The main contribution of the learning models was to show how a mimicry bias can drive the patterns of results we find, rather than differences in, for example, learning rates. The decision model then shows, by taking both choices and response times into account, how changes in the amount of evidence required for a response (“boundary separation”) could help explain why participants performed best in the congruent smile condition.

We now foreground the modeling more under Instrumental avoidance learning, Page 7, First Paragraph following the passage the Reviewer quoted, we also highlight the exploratory nature of the modeling work under The current study, Page 10, First Paragraph.

3. This is a somewhat more minor point – I also wondered a bit about how well the incongruent condition actually fits with the overarching framing. The authors describe the problem at hand like so: "For example, not realizing that an interlocutor’s smile has changed from signaling friendliness to cold politeness when engaging in casual banter can lead to social gaffes or damaged relationships." That’s very true and presents an interesting question! From an ecological validity standpoint, this isn’t what the incongruent condition does though. The right analogue would be a person who smiles when they’re happy and frowns when they’re sad.

We thank the reviewer for this piece of feedback. We have now rephrased the aforementioned sentence as "For example, not realizing that an interlocutor’s smile signaling friendliness suddenly changed to an angry frown signaling hostility when engaging in casual banter can lead to social gaffes or damaged relationships." We have now added the reviewer’s example in the Instrumental avoidance learning, Page 6, Paragraph 2

4. A bit more information regarding the stimuli would be helpful. The authors write “Twelve video clips were retrieved from the Amsterdam Dynamic Facial Expression Set (ADFES) (van der Schalk et al., 2011)” on page 10. That being said, what were the selection criteria? Were these stimuli normed in terms of expression prototypicality and intensity? Were they equated across stimuli within gender? Were they equated between stimuli *across* gender? Given the differences between the smiling and frowning expressions (and the integration of participants’ EMG responses into the feedback), it would of course be an issue if the frowns were less intense

or less prototypic than the smiles.

We did not have any explicitly defined criteria as to how to select the facial stimuli for our study. In regards to prototypicality and intensity, all the facial stimuli within the ADFES dataset were aimed primarily to capture the “core” action units (AUs) for the respective expressions (smile and frown) within the FACS system. The facial stimuli within ADFES were not equated within nor across gender, because we were not aware of any previous research arguing for the influence these effects. Furthermore, we would like to stress that we controlled for target stimulus identity in our study as a potential confound in our mixed models by adding the target stimulus identity as fixed effects analyses. These results are presented in our revised Supplementary materials, Supplementary table 19.

Also, I wasn't totally sure I had the following details right... the authors pulled three target identities within gender (e.g., 3 male, 3 female) in total from the larger set, but each individual participant only saw *two* targets (each making two different expressions at different points during the experiment). Is that accurate? If so, how were the stimulus selections managed at the individual subject level?

We apologize for this lack of clarity in our manuscript. To clarify, participants only interacted with two target interactants in the main experimental manipulation. However, there was a practice/calibration phase prior to the experimental manipulation where both female and male participants interacted with a female face or male face respectively (as a function of the participants gender). Therefore, in total, all participants interacted with three male or three female target interactants. We have now clarified these misunderstandings in the revised version of the manuscript under Stimuli, Page 11, Paragraph 1.

5. The authors gender-matched targets to participants and they had a good gender balance of participants. Given these details, did they assess whether participant gender moderated any of the learning effects? Moreover, since all the targets were white, did they also restrict participant recruitment based on race? (If so, please note participant race in the Methods. If not, why not? Participant and target race have a demonstrable impact on emotion recognition and mimicry.)

We decided to not include gender as a variable in our analyses since we had no hypotheses pertaining to this variable. To clarify, we did not want to overfit our mixed models (we did not want to add too many fixed effects without having clear hypotheses in order to justify their inclusion). Furthermore, we did not observe any effect of gender as revealed by our mixed effects model as shown in Supplementary Materials, Supplementary Table 20.

We did not restrict participants' participation on the basis of ethnicity/race. We have now clarified this piece of information under Participants, Page 10, Paragraph 1. We had no a priori hypotheses pertaining to ethnicity/race, hence we did not seek IRB permission to classify our participants accordingly. In Sweden, participant ethnicity/race is viewed as sensitive personal information and requires careful justification whenever included in a study.

6. There are number of sections in the manuscript that could really use a second read-through and careful revision. For example, there's some important detail in the following excerpt from the Methods, but some of it is very hard to parse:

“Six of the video clips consisted of three different male faces and the female faces where gender matched to male participants. Each male face had a subset of two video clips whereas is one of them, he expressed happiness and in the other video clip, expressed anger upon presentation.”

We thank the reviewer for allowing us to adjust these erroneous sentences. We have now changed it to “Six of the video clips consisted of video presentations of the faces of three different male individuals. Each male individual appeared in two video clips, expressing a smile and a frown, respectively.” These changes have been added under Materials, Page 11, Paragraph 1.

Here's another one right in the opening on page 3: “Therefore, an important question is to determine what the mechanisms underlying the learning of, and deciding about, our facial behaviours during interactions in a threatening environment are?”

We have now changed it to “Therefore, an important question is to determine what underlying mechanisms influence the learning process of our facial expression selection during interactions in a threatening environment.” under Introduction, Page 4, Paragraph 2.

Finally, there are a fair number of subject/verb agreement issues (especially in the Intro; e.g., “spontaneous facial mimicry facilitate congruent exchanges of facial expressions,” “such as when an individual learn to frown,” etc.) that should be revised.

We apologize once again for these unclarities, we have now revised the subject/verb agreement issues throughout the entire Introduction.

7. Lastly, on page 31, the authors write, “Additionally, we observed that higher ESF (Expression sub-facet of ERQ) predicted higher CR, suggesting that suppressing one's own emotional expression could facilitate a better performance.” I could be missing something, but I couldn't find this detail in the main text; it seems like it would be in the section on page 22 but isn't. (At the same time, in that section, they refer to an effect of STAI but the associated stats don't seem to represent a statistically significant result: “Finally, we observed a main effect of STAI-T ($B = 0.75$, $SE = 0.14$, $p = 0.34$) where increased STAI-T predicted lower CR.” I'm not sure what's going on here, though I may be missing something.)

We apologize for these errors and we thank the reviewer for patiently pointing these issues out. We have now removed the claim regarding ESF under Discussion, Page 28, Second Paragraph. Additionally, we have now corrected the p-value of the STAI-T (it was supposed to be 0.034 and not 0.34 as a typo occurred) under Correct response and individual differences measures, Page 21, Paragraph 1. Furthermore, we have corrected

the sentence “Finally, we observed a main effect of STAI-T where increased STAI-T predicted *lower* CR” to “We observed a main effect of STAI-T where increased STAI-T predicted *higher* CR” under Correct response and individual differences measures, Page 21, Paragraph 1 as this was a typo as well due to confusing results. A revised discussion of the STAI-T results from our mixed effect models as well as RL model have now been added under Discussion, Page 29, Paragraph 1.

Reviewer: 2

Comments to the Author(s)

The manuscript presents a novel method to examine how humans learn to produce adaptive facial responses to the facial expressions of others. Participants viewed videos of smiling or frowning faces and learned which facial response (either congruent or incongruent) are associated with avoidance of electrical shock. Results showed quicker learning with congruent facial responses, particularly smiling. The manuscript concludes that the presented method could be used to further knowledge social interaction processes.

The manuscript presents a promising method and initial results that demonstrate its potential to advance understanding of complex social interaction processes. The manuscript is very well written, and the experimental work is thorough and well considered. Overall, the manuscript presents a convincing demonstration of the method and highlights well the new research avenues that could be explored with this method. To better support this aim, certain aspects of the manuscript could be strengthened, as outlined below.

We are very happy that the reviewer thinks our manuscript was ‘very well written’, and the experimental work ‘thorough and well considered’.

The method is designed primarily for the study of human social interactions, which, as the manuscript describes well, is complex. However, the task used here is maximally simplified – i.e., a 2AFC A-not-A facial response task. A first issue is whether this simplified task is too easy, particularly in a trial-and-error situation. For example, on the very first trial, the participant only needs to choose one facial response at random to learn what the correct response is. A second issue is whether this 2AFC-based method, including the computational modelling components, can be generalized to higher N-AFC tasks (e.g., see DeCarlo, 2012, Journal of Mathematical Psychology) or where participants can freely choose which facial response to make (i.e., an unconstrained trial and error design).

We agree that our task represents a simplification of the complexities of real social interactions. Our aim was to introduce a novel method of studying facial expression decisions and that such decision-making generalizes to changes in punishment contingencies. In order to increase task difficulty, we included a Reversal phase that reversed reinforcement contingencies half-way through the experiment. Importantly, despite being a relatively simple decision situation, participants varied in their ability to accurately perform the task and we were able to observe average differences in performance depending on Congruency and Expression conditions. This suggests that our

strategy of choosing a simple task was justified for this first exploration of our paradigm, but – as the reviewer suggests – future work should scale up the complexity of the experimental situation.

Finally, we certainly agree with the reviewer that in a truly naturalistic social settings, N-AFC tasks would potentially have superior ecological validity. However, since we wanted to establish a proof of concept of modelling facial expression exchange, we had to restrict ourselves to a simplified paradigm in order to assess whether this would be feasible in future studies. Nevertheless, even if people have a large range of expressions to choose from, given the prevalence of mimicry effects, it is possible that their first decision is - like in our learning models – to copy or not copy the partner (similar to a Go - No Go decision).

In the revised manuscript, we now outline the possibility of expanding our paradigm in future studies with multiple (or unconstrained) choice alternatives under Summary, limitations, and future directions, Page 30 Paragraph 1.

The main aim of the task is to measure the facial responses of participants in response to videos of others' facial expressions. However, it's not clear whether facial responses were recorded during the videos or for how long facial responses were recorded. For example, a small involuntary facial movement could be made initially followed by the facial movement the participant intends to make. Which would be considered for recording of correct vs incorrect trials? A related question is whether certain facial muscles are more or less under voluntary control and whether the speed of activation differs.

The facial responses were indeed recorded during video/target stimulus presentation. We certainly agree with the reviewer that involuntary facial movements may have been formed before reaching the final decision since some facial muscles are governed by involuntary control.

To further clarify, in order to exclude data points where involuntary facial movements may have triggered a nonsensical outcome, we first excluded trials with a response time below 0.626s, because these instances were judged to indicate noncompliance with the task. Furthermore, we also excluded overly slow responses where participants responded later than 5s of target stimulus presentation in order to exclude outlier data points for our drift diffusion model (DDM models typically have issues processing these data points). So, within the time frame of 0.626s-5s, the target stimuli always formed and expression. The process of this exclusion criteria is explained in under Data Analyses, Pages 15-16 (Paragraph 2 in Page 15 to Paragraph 2-3 in Page 16 under Data Analyses).

Some discussion points could be developed. Specifically, learning is quicker with congruent facial expressions, particularly for smiling. Might this reflect frequency of such responses? For example, most interactions are friendly with smiling being met with smiles in return and threat-related facial expressions being rarer.

We thank the reviewer for pointing this out. Yes, reciprocating smiles vs. frowns through existing learned habit is indeed likely to explain some proportion of the variance as to why participants showed enhanced CR in the Congruency and Expression conditions.

Furthermore, as we replied to Reviewer 1 (Question 2, Page 2) facial mimicry is more frequent and consistent toward smiling faces vs. frowning faces according to the social-contextual views of emotional mimicry (Keltner and Haidt, 1999, Hess & Fischer, 2013, Parkinson, 2011) since reciprocating frowns is more dangerous than reciprocating smiles in most situations. Hence, we believe that social norms which influence friendly/unfriendly social dispositions can have influenced these results. A brief discussion of this has been added under Discussion, Pages 27-28 (from Paragraph 4 in Page 27 to Paragraph 1 in Page 28 under Discussion).

To strengthen claims that this method could be used to better understand social interactions, it would be useful to report the pattern of results per participant – that is, does the pattern of results reported replicate across participants (vs reflects an averaging artefact, see also Grice et al., 2020 Advances in Methods and Practices in Psychological Science)?

We have now added this under Supplementary Materials, Page 18, Within participant consistency, however, we would like to point out that we did not have precise hypotheses in regards to what would constitute a participant behaving according to these hypotheses. For example, 91% of the participants had higher CR on Congruent vs. Incongruent trials but they could have been anywhere from slightly better or much better on Congruent vs. Incongruent trials.

Minor points:

- Report the visual angle, in degrees (horizontal and vertical), of the face stimuli in each experiment

Each face was shown in a face-forward angle, where the target stimulus directly faced the participant.

- Explain why the face stimuli were gender matched to participants?

We wanted to minimize any potential confounding effects due to inter-gender interactions. We have now included this perspective under Stimuli, Page 11, Paragraph 1.

- Provide a rationale for the threshold of 0.001mV/s

Because our method is entirely novel, and established standards for setting the threshold was lacking, we carefully piloted several thresholds before launching this study and we determined that this threshold was adequate in order to capture an active response from the participant to the interactant. Furthermore, the validity of the threshold was strengthened by the fact that we replicated numerous previous findings within the realm of instrumental learning such as increased CR as a function of trial. Additionally, we also replicated basic results within the visuomotor literature such as enhanced CR and faster RT on congruent trials.

- Mean RTs are computed though it's not clear that the underlying data meet the assumptions of parametric analyses

We logarithmized RT because the distribution of this variable since is typically right skewed (Ratcliff 1993). To clarify, RT was logarithmized in all analyses when being used as a dependent variable and as a fixed effect.

Furthermore, we have included figures including median RT and median absolute deviation (mad) in the Supplementary Materials, Page 19, Figure S6 and S7.

- By "Caucasian" do the authors mean "white"?

Yes, the "Caucasians" were "white". We now clarify our meaning the first time "Caucasian" is used in the manuscript under Stimuli, Page 11, Paragraph 1 as well as Discussion, Page 30, Paragraph 2.

- Given that the ethnicity of stimuli and participant could impact the results, report both. **Our stimuli material did not provide the specific ethnic identity of the actors, however as mentioned above, they were "white". Furthermore, we did not restrict participants participation on the basis of ethnicity/race. we have now clarified this piece of information under Participants, Page 10, Paragraph 1. We had no a priori hypotheses pertaining to ethnicity/race, hence we did not seek IRB permission to classify our participants accordingly. In Sweden, participant ethnicity/race is viewed as sensitive personal information and requires careful justification whenever included in a study. We do, however, consider the inclusion of participant race a worthwhile endeavor in future studies.**

Similarly, report the cultural background of participants

We did not record the cultural backgrounds of the participants; we now note this limitation under Summary, limitations, and future directions, Page 30, Paragraph 2.

- The section Materials, Stimuli could be more clearly written/expressed

- "Males" and "females" or "men" and "women"

We have now consistently adjusted these sections to use the terms "males" and "females".

- Page 11, line 29/30 "clips of a face" (not "on a face")

This error has now been corrected under Procedure, Page 12, Paragraph 2.

- "Our" species could be better described as "the human" species

This error has now been corrected throughout the manuscript.

- See Zhan et al., (2019) NHB for parametric control of face features

We thank the reviewer for suggesting this paper for future references.

===PREPARING YOUR MANUSCRIPT===

- one version identifying all the changes that have been made (for instance, in coloured highlight, in bold text, or tracked changes);
- a 'clean' version of the new manuscript that incorporates the changes made, but does not highlight them. This version will be used for typesetting if your manuscript is accepted.

===PREPARING YOUR REVISION IN SCHOLARONE===

Please ensure that you include a summary of your paper at Step 2 'Type, Title, & Abstract'. This

should be no more than 100 words to explain to a non-scientific audience the key findings of your research. This will be included in a weekly highlights email circulated by the Royal Society press office to national UK, international, and scientific news outlets to promote your work.

<https://royalsociety.org/journals/authors/author-guidelines/#supplementary-material> to include a suitable title and informative caption. An example of appropriate titling and captioning may be found at https://figshare.com/articles/Table_S2_from_Is_there_a_trade-off_between_peak_performance_and_performance_breadth_across_temperatures_for_aerobic_s_cope_in_teleost_fishes_/3843624.

At Step 7 'Review & submit', you must view the PDF proof of the manuscript before you will be

able to submit the revision. Note: if any parts of the electronic submission form have not been completed, these will be noted by red message boxes.